# RF-MatID: Dataset and Benchmark for Radio Frequency Material Identification

**Xinyan Chen**[1,*] **Qinchun Li**[1] **, Ruiqin Ma**[2] **, Jiaqi Bai**[1] **, Li Yi**[2] **, Jianfei Yang**[1,†]
[1] Nanyang Technological University
[2] Ibaraki University

## Abstract

Accurate material identification plays a crucial role in embodied AI systems, enabling a wide range of applications. However, current vision-based solutions are limited by the inherent constraints of optical sensors, while radio-frequency (RF) approaches, which can reveal intrinsic material properties, have received growing attention. Despite this progress, RF-based material identification remains hindered by the lack of large-scale public datasets and the limited benchmarking of learning-based approaches. In this work, we present RF-MatID, the first open-source, large-scale, wide-band, and geometry-diverse RF dataset for fine-grained material identification. RF-MatID includes 16 fine-grained categories grouped into 5 superclasses, spanning a broad frequency range from 4 to 43.5 GHz, and comprises 142k samples in both frequency- and time-domain representations. The dataset systematically incorporates controlled geometry perturbations, including variations in incidence angle and stand-off distance. We further establish a multi-setting, multi-protocol benchmark by evaluating state-of-the-art deep learning models, assessing both in-distribution performance and out-of-distribution robustness under cross-angle and cross-distance shifts. The 5 frequency-allocation protocols enable systematic frequency- and region-level analysis, thereby facilitating real-world deployment. RF-MatID aims to enable reproducible research, accelerate algorithmic advancement, foster cross-domain robustness, and support the development of real-world application in RF-based material identification.

## 1 Introduction

Material identification is the task of identifying an object's physical category (e.g., metal, plastic) from its intrinsic properties, which is fundamental for physical artificial intelligence (Physical AI), particularly in embodied intelligence. Embodied intelligence arises from an agent's physical interaction with the environment, enabling it to perceive, reason, and adapt. Accurate material identification allows an embodied agent to infer object properties within its operational range, guiding manipulation and functional interactions. This capability has broad implications across diverse domains, such as enabling embodied agents to interpret fine-grained object attributes for physical scene understanding through material perception (Xiao et al., 2018), and to ground complex reasoning and control in material-driven functional affordances (Do et al., 2018)

Currently, material identification relies mainly on optical sensors, such as cameras and hyperspectral sensors (Drehwald et al., 2023; Xue et al., 2017; Schmid et al., 2023), to capture distinguishable spatial characteristics (e.g. texture, edges) (Erickson et al., 2020) and light spectrum features (e.g. reflectance and transmittance values) (Zahiri et al., 2022). However, vision-based material identification is inherently constrained by the visual similarity of fine-grained categories (e.g., steel vs. aluminum), limited robustness under real-world perturbations such as lighting and perspectives, and the inability of sensors to reveal intrinsic physical properties, including elasticity and conductivity.

To overcome the inherent constraints of optical sensors, non-visual sensing modalities, such as radio frequency (RF), are gaining attention as they exploit electromagnetic interactions to reveal intrinsic

---

[*]Codes are available at: https://github.com/NTUMARS/RF-MatID
[†]Corresponding Author (jianfei.yang@ntu.edu.sg)

material properties beyond surface appearance (Khushaba & Hill, 2022; Hägele et al., 2025). However, RF-based material identification has not yet been extensively explored. There are no publicly accessible large-scale datasets, hindering reproducibility and fair benchmarking across algorithms. In addition, commercial off-the-shelf (COTS) sensors offer limited and fragmented frequency coverage (Yang et al., 2024), hindering systematic evaluation of material identification across the RF spectrum, which is crucial for selecting optimal operational bands in various applications. Finally, most studies are conducted in controlled laboratory settings and rarely incorporate perturbations such as variations in sensor–object geometry (e.g., incidence angle, stand-off-distance), leaving open questions regarding robustness and scalability in deployment scenarios.

Thus, we present RF-MatID, a large-scale, wide-band, and geometry-diverse RF dataset designed to advance fine-grained material identification and enable the development of more robust Physical AI algorithms. RF-MatID encompasses 16 carefully curated fine-grained categories, systematically derived from 5 superclasses to capture subtle intra-class variations. The dataset spans a broad frequency spectrum from 4 to 43.5 GHz, sampled at 53 points per GHz. Through large-scale data acquisition, RF-MatID comprises 142k samples represented in both the frequency and time domains (71k samples in each). Note that these 142k samples correspond to representation-level instances derived from 71k unique physical measurements, each provided with a paired frequency- and time-domain representation. RF-MatID also provides 5 frequency protocols (including protocols compliant with legal frequency regulations in major global economies) and 7 split settings for a comprehensive benchmark. The key contributions and characteristics of RF-MatID are summarized as follows:

- **First open-source, wide-band, geometry-diverse RF dataset for fine-grained material identification.** To the best of our knowledge, we construct the first large-scale open-source RF dataset covering 16 fine-grained material categories from 5 superclasses. The dataset spans a wide frequency band (4–43.5 GHz), and systematically incorporates variations in incidence angle and stand-off distance to emulate realistic geometric conditions. Both time-domain and frequency-domain representations are provided, enabling multi-perspective evaluation.

- **Investigation of RF data representations and protocol-level applicability.** We evaluate RF data representations by comparing classification accuracy on raw frequency- and processed time-domain signals, showing that raw frequency-domain data can be directly leveraged by deep learning models without additional domain transformation. Additionally, we explore RF-based material identification under various frequency-allocation protocols to assess the practicality of deploying RF systems in compliance with regulatory constraints. We further evaluate consecutive sub-bands of different bandwidths to gain insights into frequency selection across diverse applications.

- **Benchmark of learning-based approaches and robustness.** We establish extensive benchmarks of state-of-the-art deep learning models on the RF material identification task, adapting architectures from computer vision, natural language processing, and RF sensing. Beyond in-distribution accuracy, we present a systematic evaluation of out-of-distribution robustness in RF sensing, through cross-angle and cross-distance domain shifts that emulate geometric perturbations.

## 2 RELATED WORK

### 2.1 MATERIAL IDENTIFICATION

Material identification has been extensively studied due to its importance in industrial (Johns et al., 2023) and civil (Ha et al., 2018) contexts in the previous decades. In the signal processing domain, physics-based approaches have been well studied for material identification (Wolff, 1990; Wu et al., 2020; Rothwell et al., 2016; Sahin et al., 2020), which employ signal processing techniques such as the Fresnel equations (Fresnel, 1834) and the Nicolson–Ross–Weir (NRW) method (Nicolson & Ross, 2007) to estimate intrinsic material electromagnetic properties (e.g. permittivity, conductivity, and absorption characteristics) for materials distinguishing. However, physics-based approaches often rely on idealized assumptions and hand-crafted formulas, introducing challenges such as sensitivity to environmental conditions, limited adaptability to diverse application scenarios, and susceptibility to extraneous object-specific factors unrelated to the material(e.g. thickness and size).

With the advancement of deep learning techniques, learning-based approaches provide more accurate and generalizable solutions for material identification. Vision-based approaches constitute the predominant paradigm for material identification. These methods exploit visual features, such as

| Dataset | Modality | Frequency (GHz) | Band-Width | # Super-Classes | # Sub-Classes | Geometric Variations | # Samples | Public Accessibility | # Benchmark Models |
|---|---|---|---|---|---|---|---|---|---|
| Ha et al. | RFID | 0.5-1.0 | 0.5 | 1 | 16 | distance, angle | 2,048 | ✗ | 4 |
| Wang et al. | | 0.92-0.93 | 0.01 | 2 | 16 | distance, angle | 1,800 | ✗ | - |
| Feng et al. | Wi-Fi | 5.0-5.8 | 0.8 | 1 | 10 | distance | 2,000 | ✗ | - |
| Shi et al. | | 2.4-2.5 | 0.1 | 4 | 14 | - | 1,568 | ✗ | - |
| Dhekne et al. | UWB | 3.5-5.5 | 2.0 | 1 | 33 | - | 330 | ✗ | - |
| Zheng et al. | | 6.5-8.0 | 1.5 | 2 | 4 | distance | 14,000 | ✗ | 3 |
| Wu et al. | mmWave | 58.7-61.3 | 2.6 | 5 | 21 | distance | 67,200 | ✗ | - |
| He et al. | | 77.0-81.0 | 4.0 | 5 | - | angle | 200,000 | ✗ | 2 |
| Shanbhag et al. | | 77.0-81.0 | 4.0 | 7 | 23 | distance, angle | 15,386 | ✗ | 3 |
| Chen et al. | | 77.0-81.0 | 4.0 | 6 | - | distance, angle | 129,600 | ✗ | 5 |
| **RF-MatID** | UWB-mmWave | 4.0-43.5 | 39.5 | 5 | 16 | distance, angle | 142,000 | ✓ | 11 |

Table 1: Comparisons of RF-MatID with other RF material identification datasets. Previous datasets are organized by sensing modality and a '–' in the table denotes the absence of corresponding dataset feature in the research work.

color, texture, and reflectance properties, allowing computer vision models to learn hierarchical representations that reveal the underlying material patterns. Driven by the ubiquitous deployment of optical sensors, numerous material image datasets and benchmarks have been established (Weinmann et al., 2014; Bell et al., 2015), enabling standardized and reproducible research and improving model accuracy and generalizability (Drehwald et al., 2023; Xue et al., 2017). Beyond RGB-based solutions, other studies have explored more physically informative optical modalities: for instance, TOF cameras capture a combination of surface and subsurface scattering effects (Su et al., 2016), while hyperspectral imaging records a full spectral profile at each image pixel (Salas et al., 2025). However, the weak penetration of near-infrared camera signals, which are typically in the MHz range, limits their ability to capture information beyond an object's surface. As a result, vision-based material identification is highly susceptible to variations in lighting and object geometry, which has spurred greater interest in robust RF-based approaches.

Existing RF-based deep learning approaches for material identification can be broadly divided into feature-based (two-stage) methods and end-to-end learning methods. Feature-based methods extend traditional physics-driven approaches by applying machine learning classifiers on manually engineered signal features for material identification (He et al., 2022; Shanbhag et al., 2023). While such classifiers can capture latent feature patterns beyond hand-crafted rules, their performance remains constrained by the quality of engineered features and shows limited adaptability to diverse real-world conditions. To overcome the limitations, recent works explore end-to-end deep learning frameworks that directly operate on RF signals, enabling discriminative representation learning of material characteristics without explicit feature extraction. (Zhang et al., 2024; Hägele et al., 2025).

## 2.2 RADIO-FREQUENCY MATERIAL DATASETS

Currently, most RF-based material datasets are collected using wireless COTS sensors operating in the 0.9–81 GHz frequency range, capturing electromagnetic signals that encode material properties (Chen et al., 2024). Accordingly, these datasets can be broadly categorized based on the sensing modality into four clusters. **Radio-Frequency-Identification (RFID)-based** datasets are mostly collected in the 0.90–0.927 GHz frequency band, and the target object is either tagged on the surface or placed between a reader and tags deployed in the environment (Wang et al., 2017; Ha et al., 2020). The modality enables applications such as through-wall sensing via strong low-frequency penetration, but the reliance on physical tags greatly complicates data collection. **Wi-Fi-based** datasets primarily leverage the 2.4 GHz and 5 GHz frequency bands, and the target object is typically placed between a Wi-Fi transmitter and receiver (Yang et al., 2023; Feng et al., 2019; Shi et al., 2021). The ubiquity of Wi-Fi devices facilitates convenient data collection, but the narrow bandwidth and distortions induced by hardware imperfections constrain the reliability of signals. **Ultra-WideBand (UWB)-based** datasets are mostly collected using ultra-wideband pulse signals in the 3.1–10.6 GHz frequency range and the equipment setup is similar to WiFi-based collection. UWB-based systems can capture fine-grained attenuation and delay properties (Dhekne et al., 2018; Zheng et al.,

2021), but face practical barriers from strict synchronization demands and high deployment costs. **Millimeter-Wave (MmWave)-based** datasets are mainly collected using COTS devices operating in the 57–64 GHz and 76–81 GHz frequency bands (Wu et al., 2020; Shanbhag et al., 2023; Chen et al., 2025). A single mmWave radar unit is an integrated transceiver that provides accurate phase information characterizing the dielectric properties of the materials. However, its limited penetration confines the sensing to material surfaces, making it highly sensitive to environmental occlusions.

However, we observe that none of the existing datasets are publicly accessible, resulting in a lack of standardized state-of-the-art comparisons across the latest algorithms. Moreover, due to limitations of COTS sensors, each dataset typically covers only a narrow frequency band, lacking diversity in the frequency domain. In addition, most datasets lack systematic benchmarks to evaluate the performance of modern deep learning models on RF-based material identification tasks. To address these gaps, we propose **RF-MatID**, the first open-source dataset and benchmark covering 39.5 GHz UWB-mmWave (Zhang & Pan, 2013) frequency band, including 16 fine-grained material categories, and providing comprehensive benchmarks with 9 models, 5 evaluation protocols, and 7 data split settings. RF-MatID is critical for advancing machine learning research in RF-based material identification and can facilitate developments in embodied AI for tasks such as indoor scene understanding, precise robotic manipulation, and affordance learning.Table 1 summarizes both previously used RF-based material datasets and our proposed RF-MatID dataset.

# 3 PRELIMINARIES OF RF SENSING

## 3.1 RF DATA PROPERTIES

Various RF-based material sensing systems, summarized in section 2.2, can be broadly categorized into radar-based and non-radar-based approaches. Benefiting from coherent transceiver design, radar-based sensing provides high-resolution amplitude and phase information, that can serve as discriminative features for learning-based material classification in indoor embodied AI tasks. Leveraging these advantages, we establish a UWB-mmWave sensing platform designed to drive real-world applications. In RF-MatID's mono-static sensing system, electromagnetic waves are transmitted from an antenna, interact with the material subject, and reflect back to the antenna. When these waves encounter a material, a portion of their energy is reflected at the surface, while the remainder is transmitted into the material. The material's intrinsic physical properties, such as permittivity and conductivity, will affect the amplitude, phase, and temporal characteristics of the electromagnetic waves. These variations form informative latent features for machine learning models, enabling fine-grained material identification. Furthermore, the behavior of electromagnetic waves is influenced by its frequency: lower-frequency waves penetrate deeper and reveal bulk properties, whereas higher-frequency waves are more sensitive to surface details but attenuate more in lossy materials. Thus, our system collects signal data spanning both the centimeter-wave band (3–30 GHz) and the millimeter-wave Q-band (30–50 GHz), ensuring complementary information capture for robust learning across diverse materials.

In our RF-based material sensing setup, the acquired data consists of complex signals uniformly sampled across the 4–43.5 GHz band. The spectrum is discretized into 2,048 frequency bins, each corresponding to a distinct carrier frequency. As illustrated in the formula below, the response at each frequency $f_i$ is represented by its in-phase ($I$) and quadrature ($Q$) components, forming a complex value.

$$H(f_i) = I(f_i) + jQ(f_i) = |H(f_i)| \, e^{j\angle H(f_i)} \tag{1}$$

Here, the magnitude $|H(f_i)|$ encodes amplitude attenuation and the phase $\angle H(f_i)$ encodes the propagation delay introduced by the material. These complementary features form the raw sensor measurements, but standard deep learning backbones operate in the real domain. This discrepancy motivated the exploration of efficient RF representations, as presented in section 4.2, enabling learning-based models to effectively leverage complementary amplitude and phase information for fine-grained material identification.

## 3.2 RF TOOLS AND PLATFORM

To facilitate the data acquisition of our RF-MatID, we develop a customized RF data collection platform. As shown in Figure 1(a), we employ an RF SPIN DRH40 (RFSpin, 2024) double ridged horn

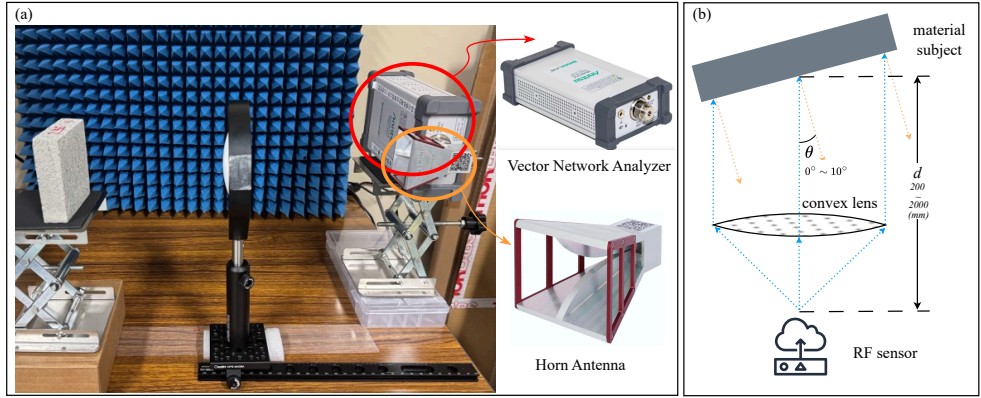

Figure 1: Data collection setup: (a) the customized sensing platform and (b) the acquisition layout.

antenna to transmit and receive signals across 4–40 GHz. The signals are subsequently processed by a 1-port vector network analyzer (MS46131A) (Anritsu, 2025), operating over 1 MHz–43.5 GHz, to generate the raw frequency-domain data. The sensor's sensing range is ∼2 m, which is intentionally tailored for indoor robot manipulation tasks in embodied AI that require high-precision, close-range perception. This design choice facilitates applications where the sensor is mounted on a robotic end-effector to enable learning and executing adaptive grasping based on material properties. In the acquisition platform, we position the RF sensor on a height-adjustable stand such that the antenna faces the material's center during data acquisition, in order to maximize signal strength and ensure that measurements capture the material's overall properties. Meanwhile, a convex lens with 15-cm-focal-length and ∼2 m sensing range is used to collimate the beam. With this configuration, the system achieves a beam footprint of 1–5 cm, representing the width of the beam on the material surface. In RF sensing, a smaller beam footprint produces a more focused beam with higher energy concentration, enabling more precise measurement of local material properties and yielding richer, more discriminative representations for learning-based models, thereby supporting fine-grained material classification. The lens ensures that the beam footprint is sufficiently small to distinguish materials of varying-sized objects under typical indoor conditions, remains robust to changes in sensing distance and background materials, and provides a consistent sensing region across samples, thereby mitigating potential biases arising from differences in material plate sizes. Appendix D.1 visualizes effective beam footprint on material plates.

## 4  DATASET

Fine-grained material identification is challenging due to subtle differences in subcategory materials, sensitivity of RF responses to geometric perturbations, and variability in the physical information captured across frequency bands. Therefore, rich data diversity in terms of fine-grained material categories, geometric perturbation simulation, and broad frequency band coverage is crucial for developing and evaluating algorithms applicable to indoor embodied AI scenarios. Thus, we introduce RF-MatID, the first large-scale, wide-band, and geometry-diverse RF dataset for material identification, containing 142k samples evenly split across dual-domain representations, with 71k in the frequency domain and 71k in the time domain. It provides 16 fine-grained subcategories, organized into five commonly encountered superclasses in indoor scenarios. To simulate the geometric perturbation from real-world data collection, samples in RF-MatID are acquired across a distance range of 200 mm to 2000 mm (at 50 mm intervals) and an angle range of 0° to 10° (at 1° intervals). Each sample spans a wide spectral band of 4 – 43.5 GHz, uniformly represented by 2,048 bins.

### 4.1  CATEGORIES OF MATERIALS

RF-MatID encompasses 16 fine-grained material categories organized into five superclasses. These superclasses represent the most common materials in indoor embodied AI scenarios: (i) bricks, (ii) glass, (iii) synthetic materials, (iv) woods, (v) stones. Within each superclass, we select multiple variants that exhibit subtle physical differences, enabling a rigorous evaluation of learning-based approaches on challenging fine-grained material classification. Specifically, the fine-grained mate-

rial categories include: for bricks, (a) overfired clay brick, (b) lightweight perforated brick, (c) lava brick; for glass, (d) transparent acrylic glass, (e) tempered glass, (f) white opaque acrylic glass; for synthetic materials, (g) melamine-faced chipboard, (h) mineral fiber board, (i) solid polyvinyl chloride sheet; for woods, (j) cedar sleeper, (k) luan plywood, (l) red oak plywood; and for stones, (m) permeable paving Stone, (n) agglomerated stone, (o) granite, (p) concrete. Figure 2 presents pictures of the 16 fine-grained materials, annotated with their corresponding tag identifiers for reference. The detailed material sample descriptions are provided in the appendix.

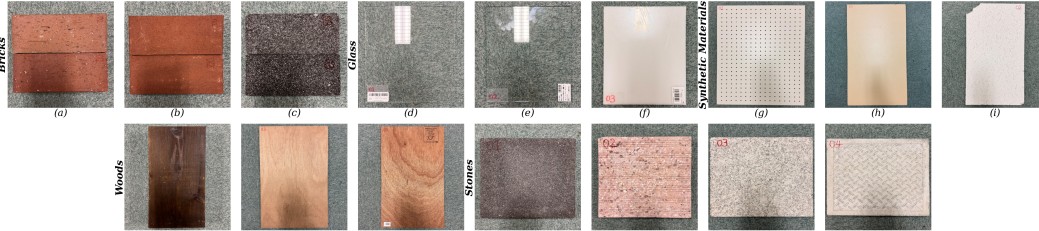

Figure 2: The visual illustration of the 16 fine-grained material categories.

## 4.2 DATA COLLECTION AND PREPROCESSING

**Realistic Data Collection** Motivated by the geometric perturbations typically encountered in data acquisition, we introduce controlled variations in both distance and incidence angle, as illustrated in Figure 1(b). The Friis transmission equation(Friis, 1946) indicates that, for fixed transmit power $P_t$, antenna gains $G_t$ and $G_r$, and wavelength $\lambda$, the received signal power $P_r$ decreases as the effective propagation path length ($r = 2d$) increases. Furthermore, for generally rough surfaces, the backscattered signal strength diminishes as the incidence angle $\theta_i$ increases relative to the incident wave, due to the cosine-dependent reduction in the backscattering coefficient $\sigma_i$, as described by the small perturbation method (El-Shenawee & Miller, 2004). In embodied AI applications, these geometric variations reflect realistic conditions where compact UWB-mmWave sensors deployed on manipulators encounter varying hand–object distances and changing incidence angles. Such variations introduce systematic changes in the RF signal representations defined in equation 1, requiring models to learn invariance and generalize across these sensing conditions.

$$P_r = P_t G_t G_r (\lambda/4\pi r)^2 \tag{2}$$

$$\sigma_i = 8k_x^4 \delta^2 cos^4\theta_i \left|\alpha_{pq}\right|^2 W(2k_x sin\theta_i), \sigma_i \propto cos^4\theta_i \tag{3}$$

Concretely, for each material, samples are systematically collected across various distances and angles configurations. Data acquisition begins at a distance of 200 mm, with an angle of 0°, where 20 samples are recorded. Each sample consists of 2048 frequency-domain bins covering the 4–43.5 GHz band. The angle is then incremented from 1° to 10°, with 10 samples collected at each angle. After completing all angles at the current distance, the distance is increased in 50 mm steps up to 2000 mm, and the same angular sampling procedure is repeated at each step. The process ensures dense coverage of the distance–angle space while increasing the sampling number at normal incidence. Appendix Figure 6 & 7 show the effect of distance and incidence angle on the frequency signal. Other forms of realistic perturbations are also discussed in the appendix B.4.

**Domain Transformation** To provide a dual-domain representation, each frequency-domain signal is paired with a time-domain signal via inverse fast Fourier transform, capturing complementary spectral and temporal features of the material response. Concretely, the 2048 ($N$) frequency bins spanning from 4 GHz ($f_{start}$) to 43.5 GHz ($f_{end}$) with uniform spacing $\Delta f = (f_{end} - f_{start})/(N-1) \approx 19.3$ MHz are converted to a 10240 ($N_t$)-length time-domain signal. Prior to transformation, the frequency spectrum is multiplied by a band-pass filter and zero-padded with $\lfloor f_{start}/\Delta f \rfloor$ leading zeros to ensure correct frequency alignment. The time resolution is $\Delta t = 1/(N_t \Delta f)$, and the time axis is computed as $z = ct/2$ with $c = 3 \times 10^8$ m/s. Both the frequency-domain and time-domain samples are saved using *comma-separated values* data format, i.e., ".csv" files.

**Frequency-Domain Representation** For frequency-domain data, a key question arises: should each frequency bin be treated as a single complex element, or should its real and imaginary parts be represented as two separate channels, i.e., a 2D ($length \times 2$) real-valued vector? To address this, we

train two models: a deep complex network (Trabelsi et al., 2017) designed for complex-valued inputs and a Bi-LSTM (Huang et al., 2015) with input dimension two for the dual-channel representation. Our experiments show that the dual-channel representation achieves higher classification accuracy, providing empirical evidence that phase information embedded in the real and imaginary parts can be effectively leveraged by learning-based frameworks.

**Post-Processing** To improve the stability and efficiency of model learning, time-domain data is standardized to have zero mean and unit variance. For the frequency-domain data in dual-channel representation, separately normalizing the real and imaginary parts would destroy the correlation (phase) information between channels. Therefore, we perform complex whitening on the combined frequency-domain data. The data is centered by subtracting its mean. A whitening transform (Koivunen & Kostinski, 1999) is then applied by multiplying with the inverse square root of the covariance matrix, obtained via eigen-decomposition. The resulting signal has zero mean, unit variance, and an identity covariance matrix, ensuring that the real and imaginary components are scaled uniformly while preserving the phase information, with its magnitude normalized.

## 4.3 INTENDED USES

RF-MatID is designed to facilitate a broad spectrum of research and applications, especially in the field of embodied AI. As the first publicly available RF material identification dataset and benchmark with diverse protocols and settings, it enables fair comparisons and accelerates the advancement of material identification algorithms. By providing a standardized dataset aligned with calibrated radar systems, RF-MatID enables the evaluation of models' transferability to real-world application scenarios. For instance, models can be trained on the 24 GHz sub-band in RF-MatID and then evaluated on radar signals captured by embodied agents operating in practical indoor scenarios, enabling assessment of their zero-shot or few-shot transferability. RF-MatID further facilitates research on domain adaptation and generalization through cross-domain evaluation settings, advancing algorithm robustness under diverse domains and environmental conditions. From a modality perspective, RF-MatID introduces a compact, low-cost sensing platform operating in the UWB–mmWave spectrum. The platform is specifically customized for indoor embodied AI applications, as it can be integrated onto a robot's end-effector to enable fine-grained local material characterization. This capability supports material aware manipulation and affordance driven workflows, such as selecting grasp strategies according to material compliance, adjusting contact forces based on surface hardness, and enabling downstream policies that rely on material grounded affordance cues, for example graspable, cuttable, or pourable objects. Moreover, the sensing platform can serve as a complementary modality within multimodal learning frameworks, providing materials' electromagnetic characteristics that enrich the information available for embodied perception (Chen & Yang, 2025; An et al., 2026). For example, when combined with vision-based perception pipelines in indoor scene understanding, the UWB-mmWave signals provide fine grained local material characteristics that can be further mapped to material grounded affordances of individual objects. When fused with visual cues that capture spatial structure and object geometry in cluttered environments, this enables embodied agents to achieve more comprehensive scene understanding across multiple dimensions, ultimately supporting more reliable reasoning and action in complex indoor settings.

## 5 BENCHMARK AND EVALUATION

In this section, we present the key benchmark configurations, the evaluation metrics, the selected deep learning models, and the baseline model design for material identification based on our proposed RF-MatID Dataset. We further evaluate experimental results to highlight the limitations of each benchmark model and demonstrate the applicability of learning-based approaches for RF-based material identification across diverse real-world scenarios.

## 5.1 BENCHMARK SETUP

**Frequency Protocol** We define five frequency-band protocols for RF-based material identification, capturing both physical distinctions (centimeter vs. millimeter waves) and region-specific regulatory constraints. Protocol 1 (**P1**) spans the full spectrum from 4–43.5 GHz. Protocol 2 (**P2**) focuses on millimeter-wave analysis, covering 30–43.5 GHz, while Protocol 3 (**P3**) targets centimeter-wave

| | | MLP | ResNet-50 | BiLSTM | Transformer | TimesNet | LSTM-ResNet | ConvNeXt | DINOv3 | Material-ID | AirTac | Baseline |
|---|---|---|---|---|---|---|---|---|---|---|---|---|
| | | Protocol 1 (4.0-43.5 GHz) | | | | | | | | | | |
| S1 | - | 99.19 | 98.85 | 86.33 | 91.76 | 99.66 | **99.84** | 99.51 | 99.28 | 96.81 | 99.77 | 99.57 |
| S2 | mod1 | 84.47 | **97.17** | 80.61 | 84.66 | 82.87 | 97.12 | 81.70 | 79.10 | 95.67 | 91.36 | 86.62 |
| | mod2 | 73.49 | 48.90 | 50.76 | 40.49 | 68.58 | 49.95 | 65.74 | 64.19 | 71.59 | **86.95** | 69.47 |
| | mod3 | 76.46 | **83.16** | 69.85 | 79.16 | 59.93 | 71.00 | 66.07 | 63.52 | 72.37 | 65.41 | 74.09 |
| S3 | mod1 | 99.08 | 99.16 | 86.78 | 87.13 | 98.59 | **99.69** | 99.45 | 98.85 | 97.63 | 98.12 | 98.89 |
| | mod2 | **90.59** | 59.48 | 77.05 | 53.38 | 78.99 | 78.13 | 85.90 | 86.27 | 51.80 | 76.60 | 85.23 |
| | mod3 | **96.39** | 90.42 | 86.50 | 82.87 | 75.50 | 80.37 | 94.84 | 95.21 | 70.54 | 75.23 | 94.21 |
| | | Protocol 2 (30.0-43.5 GHz) | | | | | | | | | | |
| S1 | - | 98.11 | **99.82** | 87.26 | 92.61 | 94.78 | 99.80 | 99.34 | 98.56 | 93.53 | 91.13 | 99.47 |
| S2 | mod1 | 88.35 | 95.86 | 83.75 | 89.35 | 84.14 | **96.82** | 87.29 | 85.13 | 93.20 | 91.16 | 86.87 |
| | mod2 | 69.19 | 58.82 | 53.31 | 56.80 | 57.96 | 50.76 | 68.90 | 62.65 | **72.43** | 71.91 | 62.31 |
| | mod3 | 79.27 | **82.12** | 77.52 | 79.64 | 75.15 | 77.02 | 81.28 | 66.81 | 75.36 | 66.89 | 79.81 |
| S3 | mod1 | 96.64 | 95.88 | 83.96 | 88.61 | 93.66 | **99.36** | 98.03 | 95.71 | 86.86 | 88.51 | 97.69 |
| | mod2 | **84.41** | 51.31 | 61.45 | 57.44 | 67.17 | 65.15 | 75.90 | 73.31 | 53.09 | 69.61 | 75.17 |
| | mod3 | 93.75 | 94.78 | 85.70 | 88.34 | 87.20 | **97.59** | 95.44 | 94.15 | 78.49 | 81.45 | 96.30 |
| | | Protocol 3 (4.0-30.0 GHz) | | | | | | | | | | |
| S1 | - | 99.52 | **99.82** | 89.87 | 88.22 | 99.64 | 99.78 | 99.35 | 98.53 | 97.32 | 98.76 | 99.47 |
| S2 | mod1 | 81.57 | 95.01 | 78.59 | 84.22 | 78.22 | 95.92 | 79.45 | 67.65 | **97.55** | 88.84 | 78.18 |
| | mod2 | 62.81 | 38.45 | 52.25 | 42.16 | 64.58 | 55.18 | **65.41** | 65.28 | 62.83 | **73.84** | 60.07 |
| | mod3 | 65.93 | 65.29 | 62.14 | **76.97** | 59.50 | 64.34 | 53.19 | 61.68 | 58.59 | 54.64 | 62.63 |
| S3 | mod1 | 99.33 | 99.31 | 89.48 | 90.14 | 98.56 | **99.51** | 99.30 | 98.30 | 98.09 | 95.51 | 99.22 |
| | mod2 | 89.79 | 79.20 | 76.28 | 62.04 | 86.26 | 78.46 | 90.28 | **90.63** | 82.40 | 87.11 | 87.17 |
| | mod3 | **95.32** | 81.63 | 87.29 | 74.63 | 72.29 | 79.34 | 93.50 | 93.23 | 65.75 | 62.77 | 93.47 |
| | | Number of Model Parameters (M) | | | | | | | | | | |
| | | 67.20 | 15.98 | 0.53 | 0.20 | 1.34 | 4.20 | 28.06 | 28.05 | 0.94 | 1.14 | 16.27 |

Table 2: Comprehensive benchmark of deep learning model end-to-end material classification performance on RF-MatID dataset. Accuracy is evaluated and shown in percentage (%) values. **Bold** indicates the best performance, while underlined values denote the second- and third-best results.

analysis, spanning 4–30 GHz. Pioneeringly, we also consider the practical feasibility of RF-based material identification under legal frequency regulations in major global economies. Protocol 4 (**P4**) covers frequency bands permitted for commercial RF sensor development in the United States, and Protocol 5 (**P5**) covers legally allowed bands in China. The details are listed in the appendix 4.

**Data Splits** To evaluate the robustness of learning-based approaches under diverse real-world conditions, we define three primary data split settings with seven sub-modes. Setting 1 (**S1: Random Split**) randomly partitions all RF signal samples into training and testing sets at a 7:3 ratio. Setting 2 (**S2: Cross-Distance Split**) partitions the dataset by distance to simulate distance-domain distribution shifts. Three sub-modes are defined: (i) **S2-1**, where 11 uniformly spaced distances (out of 37) are used as the test set; (ii) **S2-2**, where the 11 closest distances are used for testing; and (iii) **S2-3**, where the 11 farthest distances are used for testing. Setting 3 (**S3: Cross-Angle Split**) partitions the dataset by incidence angle to simulate angle-domain distribution shifts, also with three sub-modes: (i) **S3-1**, where 3 uniformly spaced angles (out of 11) are used as the test set; (ii) **S3-2**, where the 3 smallest angles are used for testing; and (iii) **S3-3**, where the 3 largest angles are used for testing. In all of the benchmark results tables, we use the term "mod" as an abbreviation for data split modes.

**Category Division** To meet various application requirements, we evaluate model performance under three category divisions. The *fine-grained division* treats all 16 materials as independent classes, regardless of their higher-level grouping. The *superclass division* groups all subclasses under each superclass into a single category (e.g., cedar sleeper, luan plywood, and red oak plywood are unified as "wood"). The *subclass division* assesses model accuracy in constrained, fine-grained tasks; specifically, we focus on the four subclasses under the "stone" category for analysis.

**Evaluation Metrics** In our benchmark, *Accuracy* is reported in all experimental analyses. To address the limitation that accuracy may be dominated by majority classes, we also report the *Macro F1-score* alongside accuracy when presenting baseline results under different task configurations. We further include precision and recall, with the detailed results reported in the appendix Table 6.

**Benchmark Models and Baseline Design** Considering the sequential dependencies across frequency bins as well as the spatial features along the frequency and channel dimensions, we benchmark a diverse set of models commonly used in computer vision, natural language processing, time-

| | | | Protocol 1 | | Protocol 2 | | Protocol 3 | | Protocol 4 | | Protocol 5 | |
|---|---|---|---|---|---|---|---|---|---|---|---|---|
| | | Settings | Acc | F1 | Acc | F1 | Acc | F1 | Acc | F1 | Acc | F1 |
| Fine-Grained Division | S1 | - | 99.57 | 99.57 | 99.47 | 99.47 | 99.47 | 99.47 | 99.53 | 99.53 | 99.41 | 99.41 |
| | S2 | mod1 | 86.62 | 86.66 | 86.87 | 86.82 | 78.18 | 78.21 | 87.35 | 87.26 | 84.82 | 84.65 |
| | | mod2 | 69.47 | 68.46 | 62.31 | 62.12 | 60.07 | 59.81 | 66.27 | 65.89 | 67.18 | 66.45 |
| | | mod3 | 74.09 | 73.07 | 79.81 | 79.22 | 62.63 | 62.67 | 78.40 | 77.46 | 72.04 | 70.63 |
| | S3 | mod1 | 98.89 | 98.89 | 97.69 | 97.69 | 99.22 | 99.22 | 98.42 | 98.42 | 99.37 | 99.38 |
| | | mod2 | 85.23 | 85.12 | 75.17 | 75.10 | 87.17 | 86.96 | 74.94 | 74.51 | 85.11 | 84.87 |
| | | mod3 | 94.21 | 94.16 | 96.30 | 96.27 | 93.47 | 93.44 | 96.65 | 96.63 | 95.95 | 95.90 |
| Superclass Division | S1 | - | 99.82 | 99.82 | 99.78 | 99.78 | 99.82 | 99.82 | 99.85 | 99.85 | 99.87 | 99.87 |
| | S2 | mod1 | 91.15 | 90.85 | 94.58 | 94.43 | 91.15 | 90.85 | 93.50 | 93.30 | 94.00 | 93.98 |
| | | mod2 | 87.24 | 86.58 | 88.54 | 87.98 | 86.05 | 85.86 | 84.69 | 84.09 | 85.19 | 84.61 |
| | | mod3 | 87.23 | 87.50 | 89.54 | 89.51 | 86.48 | 86.46 | 89.31 | 89.67 | 87.34 | 87.61 |
| | S3 | mod1 | 99.71 | 99.69 | 99.34 | 99.30 | 99.44 | 99.40 | 99.18 | 99.15 | 99.69 | 99.68 |
| | | mod2 | 91.26 | 90.73 | 89.08 | 88.64 | 94.29 | 93.96 | 85.80 | 84.74 | 91.68 | 91.09 |
| | | mod3 | 98.11 | 98.10 | 99.06 | 99.02 | 97.94 | 97.87 | 98.99 | 98.96 | 98.02 | 98.01 |
| Subclass Division | S1 | - | 99.61 | 99.61 | 99.72 | 99.72 | 99.68 | 99.68 | 99.66 | 99.66 | 99.72 | 99.72 |
| | S2 | mod1 | 98.77 | 98.77 | 96.36 | 96.36 | 95.55 | 95.51 | 96.48 | 96.45 | 98.05 | 98.05 |
| | | mod2 | 85.76 | 85.54 | 80.57 | 80.43 | 82.16 | 81.90 | 86.70 | 86.54 | 90.40 | 90.35 |
| | | mod3 | 85.81 | 85.41 | 85.80 | 85.54 | 81.67 | 81.63 | 78.75 | 76.40 | 84.15 | 83.59 |
| | S3 | mod1 | 99.17 | 99.17 | 98.81 | 98.80 | 99.10 | 99.10 | 99.55 | 99.55 | 99.05 | 99.05 |
| | | mod2 | 95.63 | 95.62 | 88.04 | 87.97 | 95.37 | 95.37 | 87.57 | 87.59 | 95.39 | 95.36 |
| | | mod3 | 98.29 | 98.29 | 97.75 | 97.74 | 98.87 | 98.87 | 99.08 | 99.08 | 98.18 | 98.17 |

Table 3: Baseline model performance under various category divisions, data split settings, and protocols. Accuracy and macro F1 score are evaluated and shown in percentage (%) values.

series, and RF-sensing research. These include Multilayer Perceptron (**MLP**) (Gardner & Dorling, 1998), **ResNet-50** (He et al., 2016), Bidirectional LSTM (**BiLSTM**) (Huang et al., 2015), **Vanilla Transformer** (Vaswani et al., 2017), **TimesNet** (Wu et al., 2022), **Material-ID** (Chen et al., 2025), **AirTac** (Zhang et al., 2024), a hybrid **LSTM–ResNet** model (Choi et al., 2018), **ConvNeXt** (Liu et al., 2022), and the recent **DINOv3** (Siméoni et al., 2025). We also introduce a simple yet robust baseline model that leverages frequency-aware positional encoding to preserve global consistency. Parallel extractors independently capture spatial and temporal features, which are then integrated into class probabilities via an MLP fusion module. The baseline model achieves an average accuracy of 85% in all experimental configurations, while the other models perform at approximately 80%. Detailed model implementations are provided in the appendix C.4.

## 5.2 RESULTS AND ANALYTICS

**Domain Comparison** Table 4 summarizes material identification results on time- and frequency-domain signals. Under Protocol 1 for fine-grained classification, LSTM-ResNet achieves comparable performance on time-domain data to the baseline model on frequency-domain data. Given the additional effort to convert frequency signals into the time domain, and significantly higher computational complexity when using 10,240-length time-domain data, we conclude that directly learning from raw frequency-domain data is more optimized and efficient.

**Benchmark Across Models** To benchmark model performance and highlight their strengths and weaknesses in material identification, we evaluate all split settings under P1–3 using the fine-grained division, as shown in Table 2. The MLP shows consistently strong performance across configurations, but it has the largest parameter size and requires careful redesign of intermediate embedding dimensions to adapt to varying protocols. ResNet-50 and the LSTM-ResNet excel under mild domain shifts (S1, S2-1, S3-1) but degrade sharply under severe shifts (S2-2/3, S3-3). SOTA vision-based models such as DINOv3 and ConvNeXt perform competitively in challenging scenarios like S3-3. However, the low resolution of the RF data hampers the stability of model convergence, leading to suboptimal results in simpler settings. Sequence-oriented models (Transformer, BiLSTM, TimesNet) underperform in most settings and are vulnerable to domain shifts. RF-sensing models generally perform well, but still exhibit notable performance drops under certain out-of-distribution conditions. The baseline model demonstrates robustness and competitive performance in most cases.

**Quantitative Baseline Results** By evaluating the baseline model across all three category divisions, seven split settings, and five protocols, we draw several insights from Table 3. RF-based

| | | Time-Domain | | Freq-Domain | |
|---|---|---|---|---|---|
| | | Acc | F1 | Acc | F1 |
| S1 | - | 96.95 | 96.95 | 99.57 | 99.57 |
| S2 | mod1 | 85.64 | 85.29 | 86.62 | 86.66 |
| | mod2 | 71.61 | 69.60 | 69.47 | 68.46 |
| | mod3 | 74.97 | 74.32 | 74.09 | 73.07 |
| S3 | mod1 | 99.65 | 99.65 | 98.89 | 98.89 |
| | mod2 | 94.43 | 94.42 | 85.23 | 85.12 |
| | mod3 | 93.70 | 93.60 | 94.21 | 94.16 |

Table 4: Comparison between material identification performance on time- and frequency-domain signals.

Figure 3: Preliminary experiments on consecutive sub-bands with various bandwidths. Accuracy in percentage value is applied for performance evaluation.

approaches achieve an average material identification accuracy of 96.78% under S1 and S2,3-mod1 across all protocols and divisions. However, performance is significantly impacted by domain shifts, with distance shifts causing an average 16.59% drop and angle shifts a 4.02% drop. We further observe that millimeter-wave signals (P2) are more robust to distance variations, whereas centimeter-wave signals (P3) better tolerate changes in incidence angle. Interestingly, country-specific legal bands (P4 and P5) achieve performance comparable to the full spectrum (P1), demonstrating the feasibility of RF-based material identification under legal constraints without significant accuracy degradation.

**Experiments on Consecutive Sub-bands with Various Bandwidths** Band analysis offers valuable guidance for selecting optimal operating frequencies and designing tailored algorithms for specific applications. In this work, we present a preliminary exploration: as shown in Figure 3, RF signals across different bandwidths generally achieve high accuracy ($> 95\%$) on S1. However, noticeable drops appear at certain ranges (e.g., around 10 GHz and in the higher band 35–43.5 GHz), offering useful insights for selecting effective frequency ranges in RF-based material identification.

**Discussions** By analyzing the benchmark results, we highlight several challenges of learning-based material identification approaches and discuss possible solutions. Training with a standard classification loss captures only data-driven correlations, neglecting physically consistent features, which leads to instability under domain shifts and unconstrained intermediate features. Incorporating physical constraints (e.g., via PINNs) guides meaningful feature learning, improving interpretability and robustness. Additionally, standard training often overfits the source domain, limiting generalization. Domain adaptation and generalization techniques address this by aligning features or adapting parameters, enhancing cross-domain transfer and overall robustness.

## 6 LIMITATIONS AND CONCLUSION

For future improvements, we identify the following limitations in the RF-MatID dataset: First, the dataset could be expanded to include richer material variability; it does not yet cover complexities such as multi-layer composite materials or significant thickness-induced signal shifts. Second, the dataset lacks diverse environmental contexts. Future work should incorporate more complex settings, such as cluttered backgrounds or large open spaces, to account for additional multipath effects and occlusion-induced biases found in real-world scenarios. Third, the RF data is sparsely sampled across the broad frequency spectrum, and the raw complex-valued signals have low feature dimensionality. To address these issues in the next-generation dataset, we will include variables like material thickness and area to diversify samples, collect real-world data from outdoor production and construction scenarios, increase the sampling rate in application-relevant frequency bands, and leverage classical signal processing techniques to expand and enrich signal feature dimensions.

In this paper, we present the first open-source, large-scale, wide-band, and geometry-diverse RF dataset for fine-grained material identification, covering 16 fine-grained categories across the 4–43.5 GHz band with controlled variations in angle and distance, and providing both time- and frequency-domain representations. We further show that raw frequency-domain signals can be effectively leveraged by deep learning models without additional domain transformations, and we evaluate their applicability under versatile protocols. Finally, by benchmarking state-of-the-art models and systematically assessing their robustness to out-of-distribution shifts, our work provides a critical foundation for developing more reliable and physically grounded RF sensing systems.

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

APPENDIX

The appendix is organized as follows:

- Section A describes the precise role of the LLMs in this research work.
- Section B provides the dataset statistics in B.1, detailed frequency allocation protocols in B.2, descriptions of material samples in B.3, discussion on other realistic perturbations in B.4, and AI-oriented intuitions in B.5.
- Section C outlines the baseline model design in C.1, discusses model-level improvements in C.2, reports complete baseline results across all metrics in C.3, details the implementation of all benchmarking learning-based models in C.4, and benchmarks classical RF methods in C.5.
- Section D visualizes RF data where D.1 shows the size of beam footprint at difference distances, and D.2 provides visualizations of frequency- and time-domain data samples across different distances in Figure 6 and different angles in Figure 7.

## A  THE USE OF LARGE LANGUAGE MODELS (LLMS)

LLMs are used to aid and polish the writing of this paper. Specifically, they assist in refining grammar, improving clarity, and enhancing the readability of the text. No LLMs are used for the retrieval and discovery of related work, nor for research ideation.

## B  DATASET DETAILS

### B.1  DATASET STATISTICS

Table 5 presents detailed statistics of the RF-MatID dataset at various levels. Overall, RF-MatID is well balanced, containing 71,040 samples in both the frequency and time domains. At the material-category level, each fine-grained category includes 8,880 samples, while each superclass contains 8,880×(number-of-subclasses). At the distance level, 3,840 samples are collected for each distance from 200 mm to 2000 mm with a step of 50 mm. At the incidence-angle level, 23,680 samples are acquired at 0°, and 11,840 samples at each of the remaining angles from 1° to 10°.

| Total number of samples | |
|---|---|
| Frequency-domain: 71,040 | Time-domain: 71,040 |
| Samples per Material Category | |
| Bricks [(a), (b), (c)]: 8,880 * 3   ;   Glass [(d), (e), (f)]: 8,880 * 3 
 Synthetic Materials [(g), (h), (i)]: 8,880 * 3  ;   Woods [(j), (k), (l)]: 8,880 * 3 
 Stones [(m), (n), (o), (p)]: 8,880 * 4 | |
| Samples per Distance ($d$) | |
| $n_d = 3,840, d \in \{200, 250, \ldots, 2000\} mm$ | |
| Samples per Incidence Angle ($\theta$) | |
| $n_{\theta=0°} = 23,680$ | $n_\theta = 11,840, \theta \in \{1°, 2°, \ldots, 10°\}$ |

Table 5: Detailed RF-MatID dataset statistics.

### B.2  DETAILS OF REGIONAL LEGAL FREQUENCY BAND

Protocol 4 and Protocol 5 are defined in accordance with the global passive service protection requirements stipulated by the ITU Radio Regulations (RR, 2024 Edition) (ITU, 2024). According to provisions such as RR 5.340, RR 5.482, RR 5.511A, RR 5.547, and RR 5.551H, certain bands between 4–44 GHz are reserved exclusively for passive services, including Radio Astronomy, Earth

Exploration-Satellite Service (EESS) passive, and meteorological sensing. These bands are strictly protected from active emissions worldwide, and include: 10.6–10.7; 15.35–15.40; 23.6–24.0; 31.3–31.8; 36.43–36.5; 42.5–43.5 (GHz).

Protocol 4 selects its frequency bands based on the requirements of the U.S. Federal Regulations (Office of the Federal Register, National Archives and Records Administration, 2024) and the protection provisions specified by the ITU Radio Regulations. Protocol 5 follows the Radio Frequency Allocation Regulations of the People's Republic of China (2023 Edition) (Ministry of Industry and Information Technology of the People's Republic of China (MIIT), 2023) issued by MIIT, excluding bands designated for amateur use and the passive protection bands defined by the ITU(ITU, 2024). The legal frequency bands of Protocol 4 and Protocol 5 are shown in Figure 4. Valid bands are obtained by filtering out non-compliant frequencies and concatenating the remaining segments. The numerical information of these valid ranges is then paired with their corresponding data to construct the training input.

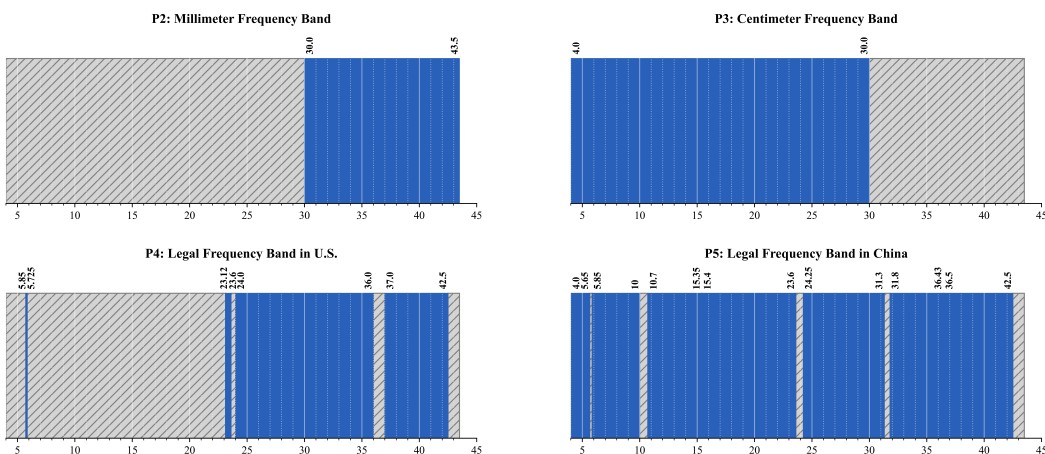

Figure 4: The visual band allocation of each frequency protocol. The frequency values are in GHz.

## B.3 DETAILED MATERIAL DESCRIPTIONS

**(a) Overfired Clay Brick**   Rectangular, $210 \times 100 \times 60$ mm, approx. 2.25 kg, made from high-iron earthenware fired above its optimal temperature. The intense firing produces a dense microstructure ($\approx 1.786 g/cm^3$) with a hard, vitrified surface ($\sim 40~\mu$m roughness), a deep red-brown body, and dispersed white grog inclusions. Its low porosity and high abrasion resistance make it suitable for high-strength masonry, paving, or other applications where durability and a rustic, speckled appearance are desired.

**(b) Lightweight Perforated Brick**   Light-weight, thin-format perforated clay brick, $210 \times 100 \times 30$ mm, extruded from fine red earthenware and fired to achieve a uniform reddish-brown body ($\approx 1.667 g/cm^3$) with a smooth, low-porosity surface ($\sim 18~\mu$m roughness). Longitudinal perforations reduce mass and thermal conductivity while maintaining dimensional accuracy and sufficient compressive strength for veneer facings, partition walls, and infill panels. Its slim profile enables rapid coursing and reduces dead load, making it well-suited for modern energy-efficient masonry and interior cladding systems.

**(c) Lava Brick**   Rectangular, $200 \times 100 \times 25$ mm, $\sim 1.08$ kg, sawn from dense vesicular basalt ($\rho \approx 2.16 g/cm^3$). Charcoal grey with a fine "salt-and-pepper" speckle of plagioclase crystals ($\sim 180~\mu$m roughness); minute sealed vesicles reduce weight while retaining strength. Fire-resistant, low-porosity, and highly abrasion-resistant, it is ideal for paving, facade cladding, or heat-storage applications.

**(d) Transparent Acrylic Glass**  The clear acrylic (PMMA) strip has $300 \times 210 \times 5$ mm size, 0.33 kg weights, $\sim 1.048 \ g/cm^3$ density. It is optically transparent, lightweight, and impact-resistant, providing a glossy, glass-like appearance without the brittleness of conventional glass.

**(e) Tempered Glass**  Rectangular tempered glass pane, $300 \times 200 \times 5$ mm, $\sim 0.72$ kg, $\sim 2.4 \ g/cm^3$ density, water-clear with the characteristic green edge tint of soda-lime float glass. Heat-toughening induces surface compressive stress, making it $3$–$5 \times$ stronger and significantly more impact- and thermal-shock-resistant than ordinary annealed glass. When broken, it shatters into small, blunt "dice" fragments, making it suitable for shelves, tabletops, appliance doors, and other applications requiring high strength and safety glazing.

**(f) White Opaque Acrylic Glass**  Solid opaque white Plexiglass, $300 \times 202 \times 10$ mm size, 0.55 kg weights, $\sim 0.908 \ g/cm^3$ density. It is lightweight and more shatter-resistant than glass. Commonly used for signage, light diffusers, photography backgrounds, shelving, and various DIY or craft projects.

**(g) Melamine-faced Chipboard**  A composite material, $600 \times 450 \times 4$ mm size, 0.87 kg weights, $\sim 0.806 \ g/cm^3$ density, consisting of a particleboard core—wood chips, shavings, and sawdust bonded with resin—covered by a thin, decorative layer of melamine or another plastic laminate. Commonly used in flat-pack furniture, including bookshelves, cabinets, and desks, for its smooth surface, durability, and ease of cleaning.

**(h) Mineral Fiber Board**  Made from a blend of mineral fibers (such as recycled slag, stone, or fiberglass), $910 \times 450 \times 10$ mm size, 0.25 kg weights, $\sim 0.061 g/cm^3$ density, fillers like perlite or clay, and a binder, typically starch. Primarily used in suspended or "drop" ceilings in commercial spaces, such as offices, schools, and retail stores, to improve acoustics by reducing echo and noise.

**(i) Solid Polyvinyl Chloride Sheet**  A solid sheet formed from a single thermoplastic polymer (PVC) via extrusion or casting. $600 \times 300 \times 9$ mm size, 0.46 kg weights, $\sim 0.284 g/cm^3$ density. It is a versatile material employed in signage, display boards, wall cladding, chemical-resistant work surfaces, and for fabricating custom parts or enclosures.

**(j) Cedar Sleeper**  Rectangular cedar sleeper, $450 \times 190 \times 95$ mm, approx. 3.6 kg (bulk $\rho \approx 0.443 g/cm^3$), sawn from knot-free heartwood with closely spaced annual rings visible on the end grain. The dark reddish-brown timber contains natural thujaplicins and other extractives that provide high durability, insect resistance, and a pleasant aromatic scent. Radial and tangential surface checks are typical as the low-density wood seasons. Ideal for landscaping borders, raised beds, or outdoor furniture where light weight, decay resistance, and a rustic, rough-sawn aesthetic are desired.

**(k) Luan Plywood**  An engineered wood panel made from thin layers of wood veneer glued together. $600 \times 300 \times 12$ mm size, 0.75 kg weights, $\sim 0.347 g/cm^3$ density. Luan is commonly used for general-purpose plywood due to its smooth surface, light weight, and affordability. It is popular for cabinetry, interior paneling, furniture backing, and various DIY projects.

**(l) Red Oak Plywood**  Made from Red Oak, a widely used and recognizable North American hardwood. $600 \times 300 \times 5$ mm size, 0.49 kg weights, $\sim 0.544 g/cm^3$ density. Its strength, durability, and attractive grain make Red Oak plywood a staple for cabinetry, furniture, flooring, and decorative interior paneling.

**(m) Permeable Paving Stone**  Square permeable paver, $300 \times 300 \times 35$ mm, approx. 3.8 kg, made by sintering angular volcanic aggregate into an open-graded matrix (bulk $\rho \approx 1.206 g/cm^3$). Dark charcoal-grey with uniformly exposed 2–5 mm basalt chips, its interconnected void network enables rapid vertical drainage while maintaining high compressive strength and freeze-thaw durability. Ideal for stormwater-friendly walkways, plazas, and green-infrastructure paving that require load-bearing capacity, slip resistance, and a rugged, monolithic volcanic appearance.

**(n) Agglomerated Stone**  A composite stone formed by binding fragments (clasts) of various rocks and minerals with a cementitious or resin-based binder. $300 \times 300 \times 30$ mm size, 6.25 kg weights,

$\sim 2.315 g/cm^3$ density. The result is a durable, uniform material that can mimic natural stone for flooring, countertops, and cladding.

**(o) Granite** A classic "salt-and-pepper" igneous rock composed of interlocking crystals of light-colored minerals (white to gray quartz and feldspar) and dark-colored minerals (black biotite mica or hornblende). $300 \times 300 \times 30$ mm size, 7 kg weights, $\sim 2.593 g/cm^3$ density. Its granular texture and high hardness make it extremely durable, widely used for countertops, flooring, building facades, and monuments.

**(p) Concrete** A man-made material without the varied crystals or patterns of natural stone. The small surface pinholes and voids are trapped air bubbles from the casting process. $300 \times 300 \times 50$ mm size, 10.65 kg weights, $\sim 2.367 g/cm^3$ density. It is valued for its versatility, compressive strength, and adaptability in construction and paving applications.

### B.4 DISCUSSION OF OTHER REALISTIC PERTURBATIONS

Beyond systematically incorporating geometric variations that reflect typical perturbations encountered in indoor embodied AI scenarios, RF-MatID also inherently captures aspects of real-world RF conditions, including material variability and multipath reflections. **Material variability** (e.g., in density, roughness, and dielectric properties) is inherently reflected in the dataset. For instance, lightweight perforated bricks and lava bricks produce distinct signatures in the raw frequency-domain signals due to materials' density and surface roughness differences. **Multipath effects** are also naturally present, as time-domain signals visualized in Appendix Figure 7 showing secondary reflections beyond the direct path.

RF-MatID intentionally does not include explicit perturbations from environmental variability, mechanical vibrations, or electromagnetic (EM) interference based on the following considerations. **Environmental factors**, such as humidity, are relatively controlled in typical indoor embodied AI scenarios and are expected to have a negligible impact on RF signal propagation and material characterization. **Mechanical vibrations** are typically compensated by the robot's control algorithms. **EM interference** has minimal effect on our FMCW radar measurements in indoor settings. The continuous linear frequency modulation of FMCW signals allows echo separation even under multi-target or overlapping frequency conditions . Moreover, commercial off-the-shelf RF devices overlap with our sensor's operating bands only in narrow frequency segments (e.g., 0.8 GHz of 5–5.8 GHz WiFi and 1.5 GHz segments of UWB bands), and our sub-band analysis demonstrates that material classification can be reliably performed outside these ranges.

### B.5 DISCUSSION OF MECHANICAL GROUNDED AI-ORIENTATION INTUITIONS

We will discuss the following AI-oriented intuitions based on mechanical material-specific details that worth future explorations.

Complex Signal Representations: Standard deep learning backbones operate in the real domain, this presents a critical architectural choice: whether to employ specialized Complex-Valued Neural Networks (CVNNs) or to project the data into a dual-channel real-valued representation. Our experiments in Section 4.2 demonstrate that the dual-channel approach is superior. It allows standard models to effectively learn the latent interactions between amplitude and phase, yielding improved out-of-distribution (OOD) generalization and more accurate fine-grained classification.

Embedded Physical Constraints: In physical science, radar equations and other physical laws are well studied. Incorporate them as regularization terms or hard constraints in the model could guide feature learning according to known material electromagnetic responses.

Disentangled Representation Learning: In material science, intrinsic material properties (e.g., permittivity, density) can be separated from geometric factors (e.g., distance, incidence angle). Incorporating disentangled representation learning could guide the model to capture geometry-invariant material features while representing geometric variations linearly.

Spectral Attention: In RF sensing, materials are identified by unique characteristics occurring at certain frequencies (e.g., periodic fluctuations or sharp energy changes) that reflect their thickness

and internal structure rather than overall signal strength. Frequency-domain attention can guide models to focus on the most informative frequency spectrum for material discrimination.

## C    BENCHMARK DETAILS

### C.1    BASELINE MODEL

To address the instability of models specializing for spatial or temporal features under complex experimental configurations, and frequency-domain data under realistic protocols is composed of multiple non-contiguous slices, we design a simple yet robust baseline model. A frequency-aware positional encoding is introduced to preserve global consistency across fragmented frequency-domain data, ensuring that an identical frequency position yields an identical embedding. To independently extract spatial and temporal features, dedicated extractors are utilized in a parallel manner. Specifically, we utilize a 3-stage ConvNext model as a spatial feature extractor and a 3-encoder-layer Transformer structure as a temporal feature extractor. Finally, a fusion module is applied to integrate features into class probability distributions. We experiment with both attention-based and fully connected fusion mechanisms, and empirically adopt an MLP-based fusion design due to its effectiveness.

### C.2    MODEL-LEVEL DISCUSSIONS

By analysing the benchmark results, we highlight several challenges of learning-based material identification approaches and discuss possible solutions. Firstly, training solely with a standard classification loss captures only data-driven correlations rather than physically consistent discriminative features, leading to instability under domain shifts and offering no constraints on intermediate feature distributions. Incorporating physical equations or consistency constraints (e.g., via PINNs) can guide the model to learn meaningful features, improving interpretability and robustness. Secondly, models trained under standard procedures also tend to overfit the source domain, limiting generalization to unseen domains. Domain adaptation and domain generalization techniques can mitigate this by aligning feature distributions or adapting model parameters, thereby enhancing cross-domain transfer and overall robustness. Thirdly, existing baseline models jointly train spatial and sequential feature extractors, limiting feature expressiveness, which can be mitigated by incorporating pre-trained modules that capture more distinctive representations. Fourthly, directly concatenating spatial and temporal features and fusing them with an MLP can lead to misaligned representations and fails to explicitly model the physical relationships between space and time, which can be alleviated by introducing feature alignment and designing a physics-inspired fusion module.

### C.3    ALL CLASSIFICATION METRICS EVALUATED IN EXPERIMENTS

We adopt four widely used metrics to evaluate model performance on the material identification task. *Accuracy*, as the most common metric, is reported in all experimental analyses. *Precision* is defined as the ratio of correctly identified positives to all predicted positives, while *Recall* is the ratio of correctly identified positives to all actual positives. Since precision and recall individually provide only a limited view of model performance, their detailed results are reported in the Table 6. To address the limitation that accuracy may be dominated by majority classes, we also report the *Macro F1-score*, which computes the F1-score for each class independently and then averages across classes.

### C.4    BENCHMARK MODELS

**LSTM ResNet:**    We pair a multi-layer LSTM front-end with a 1D ResNet back-end to capture both long-range dependencies and local motifs in TERA-MATERIAL's RF sequences. The input is a dual-channel stream (I/Q), so the LSTM ingests (T, 2) and outputs a contextualized (T, H) representation. We then permute to (H, T) and feed it to a 1D ResNet (Conv1D/BN1D/MaxPool1D with residual blocks and stagewise down-sampling), converting the LSTM's hidden size into convolutional channels. Unlike the 2D image variant, all kernels are 1D to operate along frequency/time. An AdaptiveAvgPool1D makes the model length-agnostic, and a lightweight MLP head (512→256→C)

### Fine-Grained Division

| | S1 | S2 | | | S3 | | | S1 | S2 | | | S3 | | |
|---|---|---|---|---|---|---|---|---|---|---|---|---|---|---|
| | - | mod1 | mod2 | mod3 | mod1 | mod2 | mod3 | - | mod1 | mod2 | mod3 | mod1 | mod2 | mod3 |
| P1 | 99.57 | 86.62 | 69.47 | 74.09 | 98.89 | 85.23 | 94.21 | 99.57 | 86.66 | 68.46 | 73.07 | 98.89 | 85.12 | 94.16 |
| P2 | 99.47 | 86.87 | 62.31 | 79.81 | 97.69 | 75.17 | 96.30 | 99.47 | 86.82 | 62.12 | 79.22 | 97.69 | 75.10 | 96.27 |
| P3 | 99.47 | 78.18 | 60.07 | 62.63 | 99.22 | 87.17 | 93.47 | 99.47 | 78.21 | 59.81 | 62.67 | 99.22 | 86.96 | 93.44 |
| P4 | 99.53 | 87.35 | 66.27 | 78.40 | 98.42 | 74.94 | 96.65 | 99.53 | 87.26 | 65.89 | 77.46 | 98.42 | 74.51 | 96.63 |
| P5 | 99.41 | 84.82 | 67.18 | 72.04 | 99.37 | 85.11 | 95.95 | 99.41 | 84.65 | 66.45 | 70.63 | 99.38 | 84.87 | 95.90 |
| P1 | 99.57 | 86.62 | 69.47 | 74.09 | 98.89 | 85.23 | 94.21 | 99.58 | 87.00 | 69.99 | 77.76 | 98.92 | 85.52 | 94.68 |
| P2 | 99.47 | 86.87 | 62.31 | 79.81 | 97.69 | 75.17 | 96.30 | 99.48 | 87.23 | 63.81 | 81.52 | 97.72 | 76.16 | 96.61 |
| P3 | 99.47 | 78.18 | 60.07 | 62.63 | 99.22 | 87.17 | 93.47 | 99.47 | 78.50 | 64.41 | 69.01 | 99.23 | 87.55 | 93.95 |
| P4 | 99.53 | 87.35 | 66.27 | 78.40 | 98.42 | 74.94 | 96.65 | 99.54 | 87.64 | 66.44 | 81.56 | 98.44 | 76.11 | 96.94 |
| P5 | 99.42 | 84.82 | 67.18 | 72.04 | 99.37 | 85.11 | 95.95 | 99.41 | 85.05 | 68.26 | 76.89 | 99.38 | 85.71 | 96.05 |

### Superclass Division

| | S1 | S2 | | | S3 | | | S1 | S2 | | | S3 | | |
|---|---|---|---|---|---|---|---|---|---|---|---|---|---|---|
| | - | mod1 | mod2 | mod3 | mod1 | mod2 | mod3 | - | mod1 | mod2 | mod3 | mod1 | mod2 | mod3 |
| P1 | 99.82 | 91.15 | 87.24 | 87.23 | 99.71 | 91.26 | 98.11 | 99.82 | 90.85 | 86.58 | 87.50 | 99.69 | 90.73 | 98.10 |
| P2 | 99.78 | 94.58 | 88.54 | 89.54 | 99.34 | 89.08 | 99.06 | 99.78 | 94.43 | 87.98 | 89.51 | 99.30 | 88.64 | 99.02 |
| P3 | 99.82 | 91.15 | 86.05 | 86.48 | 99.44 | 94.29 | 97.94 | 99.82 | 90.85 | 85.86 | 86.46 | 99.40 | 93.96 | 97.87 |
| P4 | 99.85 | 93.50 | 84.69 | 89.31 | 99.18 | 85.80 | 98.99 | 99.85 | 93.30 | 84.09 | 89.67 | 99.15 | 84.74 | 98.96 |
| P5 | 99.87 | 94.00 | 85.19 | 87.34 | 99.69 | 91.68 | 98.02 | 99.87 | 93.98 | 84.61 | 87.61 | 99.68 | 91.09 | 98.01 |
| P1 | 99.81 | 93.46 | 86.97 | 87.54 | 99.69 | 90.92 | 98.18 | 99.83 | 93.60 | 87.17 | 90.49 | 99.70 | 91.21 | 98.04 |
| P2 | 99.77 | 94.41 | 88.23 | 89.53 | 99.30 | 88.51 | 99.00 | 99.79 | 94.57 | 88.60 | 89.56 | 99.33 | 88.88 | 99.07 |
| P3 | 99.81 | 90.87 | 85.83 | 86.49 | 99.40 | 94.09 | 97.93 | 99.82 | 90.88 | 86.03 | 86.56 | 99.41 | 94.52 | 97.84 |
| P4 | 99.84 | 93.26 | 84.06 | 89.64 | 99.13 | 85.14 | 98.97 | 99.85 | 93.74 | 84.68 | 90.30 | 99.19 | 85.26 | 98.96 |
| P5 | 99.86 | 93.98 | 85.17 | 87.47 | 99.67 | 91.25 | 98.10 | 99.87 | 93.99 | 84.68 | 89.23 | 99.69 | 91.68 | 97.94 |

### Subclass Division

| | S1 | S2 | | | S3 | | | S1 | S2 | | | S3 | | |
|---|---|---|---|---|---|---|---|---|---|---|---|---|---|---|
| | - | mod1 | mod2 | mod3 | mod1 | mod2 | mod3 | - | mod1 | mod2 | mod3 | mod1 | mod2 | mod3 |
| P1 | 99.61 | 98.77 | 85.76 | 85.81 | 99.17 | 95.63 | 98.29 | 99.61 | 98.77 | 85.54 | 85.41 | 99.17 | 95.62 | 98.29 |
| P2 | 99.72 | 96.36 | 80.57 | 85.80 | 98.81 | 88.04 | 97.75 | 99.72 | 96.36 | 80.43 | 85.54 | 98.80 | 87.97 | 97.74 |
| P3 | 99.68 | 95.55 | 82.16 | 81.67 | 99.10 | 95.37 | 98.87 | 99.68 | 95.51 | 81.90 | 81.63 | 99.10 | 95.37 | 98.87 |
| P4 | 99.66 | 96.48 | 86.70 | 78.75 | 99.55 | 87.57 | 99.08 | 99.66 | 96.45 | 86.54 | 76.40 | 99.55 | 87.59 | 99.08 |
| P5 | 99.72 | 98.05 | 90.40 | 84.15 | 99.05 | 95.39 | 98.18 | 99.72 | 98.05 | 90.35 | 83.59 | 99.05 | 95.36 | 98.17 |
| P1 | 99.60 | 98.77 | 85.76 | 85.81 | 99.17 | 95.62 | 98.29 | 99.62 | 98.78 | 86.78 | 87.36 | 99.17 | 95.72 | 98.30 |
| P2 | 99.72 | 96.36 | 80.57 | 85.80 | 98.81 | 88.04 | 97.75 | 99.72 | 96.39 | 82.88 | 86.71 | 98.83 | 88.13 | 97.79 |
| P3 | 99.68 | 95.50 | 82.16 | 81.67 | 99.10 | 95.37 | 98.87 | 99.68 | 95.79 | 82.34 | 81.81 | 99.10 | 95.39 | 98.89 |
| P4 | 99.66 | 96.48 | 86.70 | 78.75 | 99.55 | 87.57 | 99.08 | 99.66 | 96.47 | 88.63 | 82.06 | 99.55 | 87.73 | 99.11 |
| P5 | 99.72 | 98.05 | 90.40 | 84.15 | 99.05 | 95.39 | 98.18 | 99.72 | 98.06 | 92.03 | 85.60 | 99.06 | 95.45 | 98.20 |

Table 6: Complete baseline results under various category divisions, data split settings, and protocols. Accuracy (  ), macro F1 score (  ), Recall (  ) and Precision (  ) are evaluated and shown in percentage (%) values.

performs classification. This hybrid design targets amplitude/phase order (LSTM) and fine-grained spectral patterns (ResNet), improving robustness to distance/angle perturbations.

**1D Transformer:** We treat each spectral/time bin as a token and apply a lightweight Transformer encoder tailored to 1D RF signals. A linear input projection maps multi-channel inputs (e.g., I/Q) to an embedding; a fixed sinusoidal positional encoding preserves order over the 2,048 bins. We use PyTorch encoder layers with batch-first layout and a compact feedforward width (256) plus 0.1 dropout to control capacity and mitigate overfitting under distance/angle perturbations. Instead of a CLS token, we adopt global mean pooling across tokens, which is simple, stable, and length-flexible (for any sequence less than or equal to the preset max). A single linear head produces logits. This design captures long-range cross-band interactions without imposing locality biases from convolutions.

**ResNet-50 (1D):** We adapt the image ResNet-50 to 1D RF sequences by replacing 2D kernels with 1×1–3×1–1×1 bottlenecks and BatchNorm1D. Dual-channel I/Q inputs (B, L, 2) are transposed to (B, 2, L) and passed through a 7×1, stride-2 stem with max-pooling, followed by stages [3,4,6,3] of Bottleneck1D blocks. Down-sampling is applied via stride on the 3×1 conv and a projection shortcut when shape/stride changes, preserving residual alignment. An AdaptiveAvgPool1D yields length-agnostic features, and a 2048→C linear head performs classification. This 1D design enforces

locality along frequency/time, capturing spectral edges and resonances while providing translation invariance and hierarchical abstraction; it is a strong baseline under distance/angle perturbations.

**DINOv3 (ConvNeXt-Tiny adapter):** We repurpose a DINOv3-pretrained ConvNeXt-Tiny as a feature extractor for 1D RF sequences. The sequence (T, 2) is chunked into non-overlapping patches (length = patch_size), each flattened and linearly embedded to 1024 dims, then reshaped to 32×32. These per-patch "images" are stacked along the channel axis (channels = T/patch_size) and fed to the backbone after replacing the first Conv2D to accept this channel count (kernel/stride/padding preserved). Backbone parameters can be frozen to retain DINOv3 priors. We take the backbone output, apply global average pooling (2D or token-wise), and train a single linear head. This design transfers robust DINOv3 texture/shape priors to RF, while segmenting the sequence into fixed-length patches exposes local spectral patterns and preserves long-range cross-band context through deep receptive fields.

**BiLSTM:** We use a two-layer bidirectional LSTM (hidden=128 per direction, batch_first=True) tailored to dual-channel I/Q inputs shaped (T, 2). Instead of pooling over time, we concatenate the last hidden states from the forward and backward passes, hn[-2], hn[-1], to obtain a compact 256-D sequence summary that is sequence-length agnostic. A bias-free linear head maps this summary to class logits, reducing parameters and regularizing the classifier. Bidirectionality captures cross-band dependencies that can appear in both causal and anti-causal orderings, while the final-state readout emphasizes global context over local noise from distance/angle perturbations. This minimalist design is fast, memory-light, and effective for fine-grained material discrimination.

**MLP:** As a no-frills baseline, we flatten each (2048, 2) I/Q sequence to a 4,096-D vector and feed it to a two-layer MLP with an expansion ratio of 2 (4096→8192→4096) and ReLU nonlinearities. This deliberately discards ordering, testing whether global co-occurrence statistics across the spectrum suffice for discrimination on TERA-MATERIAL. The final bias-free linear classifier maps the 4,096-D representation to class logits, slightly regularizing the head. The design creates a dense, global cross-band mixing without inductive biases from convolutions/attention, providing a parameter-efficient, GPU-friendly baseline. The trade-off is a fixed input length (tied to 2048×2); we note possible extensions (lazy init/padding/masking) to support variable lengths while preserving the simplicity of the architecture.

**TimesNet:** We adapt TimesNet for classification on RF I/Q by mapping (T, 2) into $d_{model} = 32$ with a Conv1D token embedding (kernel=3, circular padding) plus sinusoidal positional encoding—no calendar/time features. Each TimesBlock estimates the top-k periods via rFFT, pads to period multiples, folds the series into a $(steps \times period)$ grid, and applies two Inception-style 2D conv stacks (32→128→32) to capture temporal 2D-variation. Outputs from the top-k periods are softmax-weighted and added residually. We use k=3, one TimesBlock layer, and LayerNorm. For classification, we apply $GELU + dropout$, flatten $(T \cdot d_{model})$, and a linear head to C classes. These choices target multi-scale periodicity and fine spectral textures from material resonances while keeping parameters modest.

**ConvNeXt:** We adapt ConvNeXt to RF I/Q by reformatting 1D sequences into pseudo-images. A preprocessing stage uses Conv1d (kernel=stride=patch_size) to project each patch of (T, 2) into 1024 dims, adds sinusoidal positional encodings, then reshapes 1024→32×32. Patches are stacked as channels, yielding (B, $N_{patch}$, 32, 32); we set ConvNeXt's in_chans = $N_{patch}$. When T isn't divisible by patch_size, we pad to the nearest multiple. The backbone is standard ConvNeXt (7×7 depthwise conv, LayerNorm, 4×MLP, DropPath, layer scale; downsampling 4 × then 2×, 2×, 2×), followed by global average pooling and a linear head. This mapping lets spatial kernels learn intra-patch spectral texture, while pointwise mixing aggregates cross-patch cues—building robust, hierarchical features under distance/angle perturbations.

## C.5 BENCHMARK ON CLASSICAL RF METHODS

We have surveyed classical RF signal-processing and hybrid approaches specifically proposed for material identification, selected two representative methods, mSense (Wu et al., 2020) and

RFVibe (Shanbhag et al., 2023), and evaluated them under the same RF-MatID protocols. The results have been incorporated into the table below.

We observe that mSense fails to distinguish the fine-grained material categories, primarily because classical methods rely on background-only measurements for noise removal, while our dataset trains directly on signals that include background noise.

| | | mSense | | | RFVibe | | |
|---|---|---|---|---|---|---|---|
| | | P1 | P2 | P3 | P1 | P2 | P3 |
| S1 | - | 10.31 | 8.53 | 10.05 | 83.76 | 86.48 | 79.06 |
| S2 | mod1 | 9.91 | 8.47 | 10.33 | 80.48 | 82.51 | 91.09 |
| | mod2 | 13.10 | 10.25 | 10.74 | 50.36 | 52.16 | 50.28 |
| | mod3 | 8.51 | 7.95 | 8.20 | 72.20 | 82.70 | 67.93 |
| S3 | mod1 | 10.38 | 9.74 | 9.09 | 83.45 | 86.50 | 80.27 |
| | mod2 | 7.50 | 6.38 | 10.30 | 55.97 | 69.59 | 76.50 |
| | mod3 | 6.42 | 6.81 | 6.42 | 76.48 | 92.93 | 71.28 |

Table 7: Benchmark of a classical RF signal-processing approach and a hybrid method on RF-MatID dataset. Accuracy is evaluated and shown in percentage values.

# D VISUALIZATIONS

## D.1 BEAM FOOTPRINT VISUALIZATIONS

This subsection presents visualizations of effective beam footprints on material plates. In the figure below, D2 refers to the placement distance and the color bar on the right side indicates the energy concentration values.

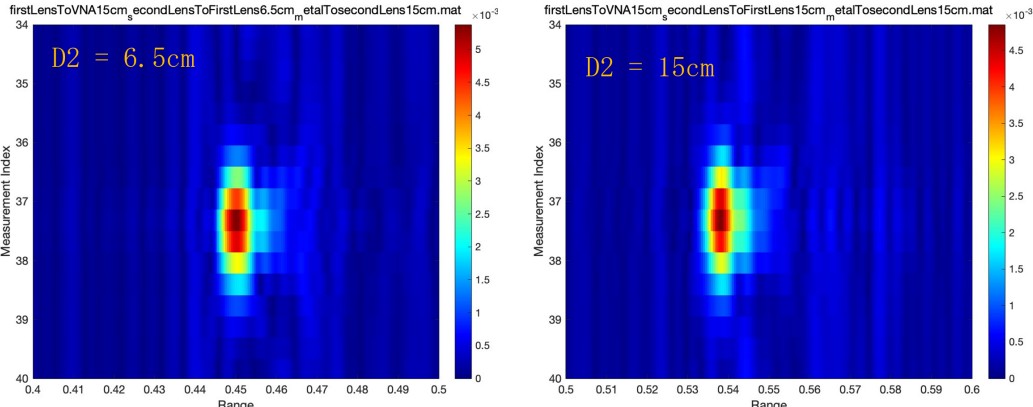

Figure 5: The visualization of beam footprints across various distances

## D.2 DATA SAMPLE VISUALIZATIONS

This subsection presents visualizations of both time- and frequency-domain samples. The frequency-domain complex signals are plotted in terms of their real and imaginary components.

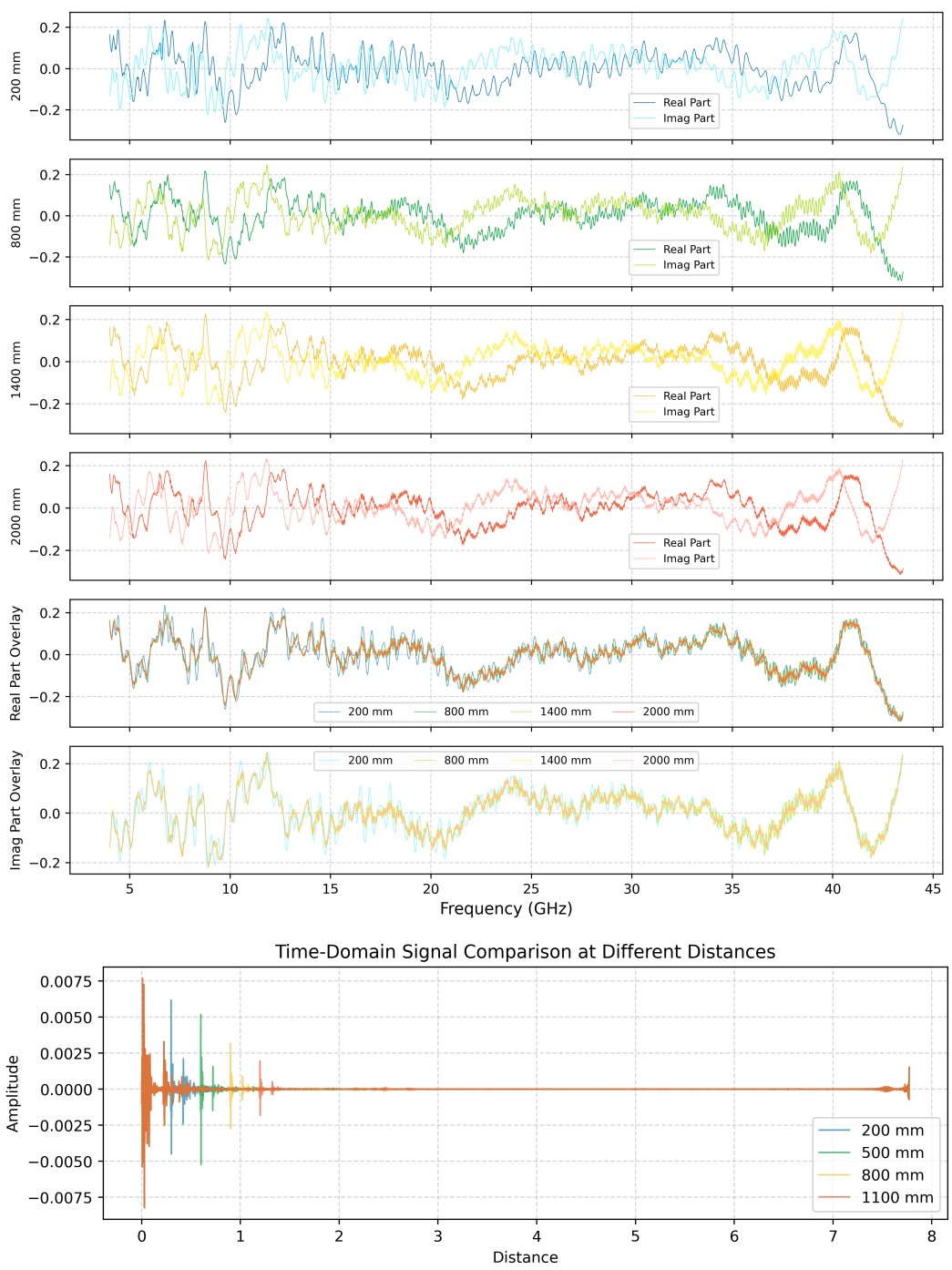

Figure 6: The visualization of frequency-domain and time-domain data across various distances

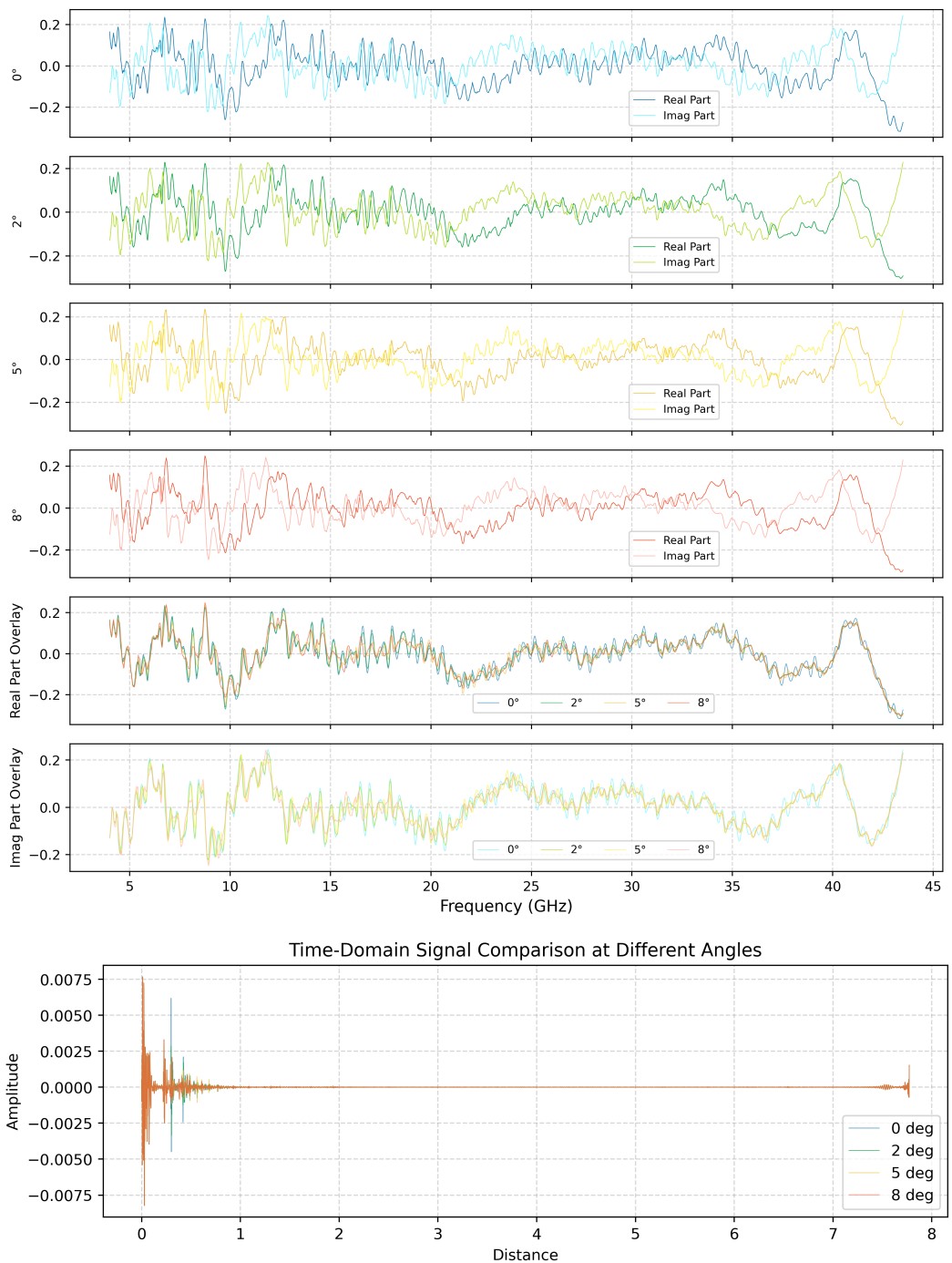

Figure 7: The visualization of frequency-domain and time-domain data across various angles

