# OpenReview forum: "RF-MatID: Dataset and Benchmark for Radio Frequency Material Identification"
_ICLR.cc/2026/Conference — ICLR 2026 Poster_

### Official Review · Reviewer_QnNM · 2025-10-27

**Soundness:** 2
**Presentation:** 2
**Contribution:** 2
**Rating:** 2
**Confidence:** 3

**Summary:**

The authors introduce a new RF-sensing based benchmark called RF-MatID for material identification. This benchmark is large in its scale across the frequency range from 4 to 43.5 GHz and the number of samples in the time/frequency domains.
The dataset contains 5 superclasses (brick, glass, synthetic materials, woods, and stones) and 16 fine-grained material categories for different variants of the superclasses. Each sample is collected at varying distances (200–2000 mm) and incidence angles (0–10°) to simulate real-world perturbations. The authors benchmark nine models popular in vision, language, and time-series domains across multiple settings, protocols, and divisions. The results have high in-domain accuracy around 99% and analyze robustness to cross-angle and cross-distance shifts. Their results show that raw frequency-domain data performs comparably or better than time-domain conversions, and that the dataset supports evaluation under different global frequency regulations.

**Strengths:**

- The dataset is clearly described and prior limitations are organized well
- The benchmark provides a good structured taxonomy for fine-grained classification.
- The benchmark covers comprehensive materials with 16 fine-grained attributes and five superclasses
- The authors evaluate the benchmark on some popular deep learning models from the vision domain.

**Weaknesses:**

- The dataset operates at a sensing range of only ~2 m, which is insufficient for real-world applications such as autonomous driving, robotics, or drone perception that typically require >10 m detection. This limitation substantially weakens the claimed motivation of supporting “autonomous systems” and confines the dataset’s applicability to laboratory-scale studies.
- The introduced perturbations are restricted to geometric variations (distance and angle) within a highly controlled indoor environment. The absence of electromagnetic interference, multipath reflections, or environmental factors (e.g., humidity, surface roughness variability) prevents the dataset from reflecting realistic RF conditions. While the authors acknowledge this limitation, the overall contribution remains narrower than claimed.
- The inclusion of both frequency- and time-domain data provides limited additional value, as the time-domain samples are directly computed via inverse FFT. This dual-domain representation described in the paper does not really introduce new empirical diversity but rather duplicates information that practitioners could easily derive themselves. The dataset nominally contains 142k samples; however, only 71k represent unique physical measurements. But the dual-domain counting somewhat inflates the dataset’s reported scale, and it is therefore concerning whether the reported dataset size is artificially inflated by redundancy and the contribution is overclaiming.
- The benchmark focuses solely on material classification. The benchmark, therefore, provides limited insight into the broader applicability of RF sensing for real-world material-based applications.
- Most of the models evaluated are vision, language, and time-series based. The baseline encodes some frequency information. However, there are many works on RF-sensing models, but they are not evaluated.
- From Figure 2, many material samples appear visually and structurally similar, suggesting a high intra-class correlation. The dataset lacks evidence of variability in properties such as texture, thickness, or surface roughness, making it unclear whether the reported generalization extends to real-world materials beyond those collected in the lab.
- **Minor:**
    - The use of the term “mod” for both *modality* and *mode* is confusing and should be clarified in the benchmark descriptions to avoid misinterpretation.

**Questions:**

Please see my weakness for most of the concerns. Some questions for authors to discuss are:
- Can the authors justify the dataset’s relevance to long-range sensing tasks such as autonomous driving or outdoor perception given that the sensing range is very close-distance
- Since the frequency-domain data can be easily transformed into the time-domain via FFT, what new information does the time-domain dataset provide that cannot be derived from the frequency data?
- Are the perturbations (distance and angle) annotated per sample, and are these metadata available in the dataset? Are there more textual annotations that we could use to give LLM the capabilities to describe the material?
- The paper reports high accuracy (>99%) across models, could this indicate potential overfitting due to low intra-class diversity or lack of environmental variability? And given the already high performance on the proposed benchmark, is the task sufficiently challenging to drive future methodological improvement?

---

> ### Author Response · Authors · 2025-11-19
> **Response to Reviewer QnNM (1/5)**
>
> We appreciate the reviewer QnNM for the valuable questions from the perspective of research method, system engineering, and practical usage of datasets. Despite many opinions, we are glad that the reviewer still acknowledges that **“the dataset is clearly described and prior limitations are organized well”** and **“the benchmark provides a good structured taxonomy for fine-grained classification and covers comprehensive materials”**. We have been working hard to improve the quality of manuscript by addressing all the concerns and adding extensive discussions for clarification. In addition, we have submitted a revised manuscript with an appendix where we mark all the suggested tables, figures, and analytics in magenta color. We hope the improved manuscript could be satisfied.
>
> Before the discussion, we would like to clarify that RF-based material identification remains largely unexplored regarding the lack of publicly accessible large-scale datasets and sensors capable of providing rich frequency-domain information. **RF-MatID is the first systematic dataset and benchmark facilitating the research direction specifically in the embodied AI scenarios**, rather than broader industrial automation applications. Moreover, commercial off-the-shelf (COTS) RF instruments cannot meet the task requirements due to fragmented frequency coverage and large beam footprints, which motivated us to design a **custom, non–off-the-shelf RF sensor** capable of collecting wide-band fine-grained material information. Consequently, the data acquisition cost is inherently high.
>
> Given these considerations, RF-MatID represents **the largest and most comprehensive dataset** of its kind, as summarized in Table 1, covering fine-grained material categories and diverse geometric perturbations through a scientifically controlled collection process. RF-MatID is not an in-the-wild dataset but a first-edition, controlled-environment dataset designed to provide a foundational and systematic study supporting RF-based material identification research in the embodied AI community. Rather than relying on COTS systems, we specifically designed a custom, non–off-the-shelf RF sensor to further establish the foundation for related research. In the future, we plan to collect larger-scale, wider-band, and more realistic-perturbation–aware datasets tailored to diverse embodied robotic applications.
>
> **Q1: How could the dataset’s ~2 m sensing range supports long-range sensing applications in autonomous systems that typically require >10 m detection?**
>
> **Answer**: RF-MatID’s ~2m sensing range is intentionally tailored for indoor robot manipulation tasks in embodied AI that require high-precision, close-range robotic perception, rather than for broader industrial or autonomous-systems applications that typically require sensing distances beyond 10m.
>
> For indoor robotic manipulation scenarios, 2-meter sensing range is sufficient for embodied agents to perceive material characteristics and further enable tasks such as adaptive manipulation and affordance learning. This is because the operational range of embodied agents is typically limited to 1–2 m, making a ~2 m sensing range sufficient for most manipulable objects and interaction settings. For example, when a robot is asked to retrieve a glass cup from a cluttered shelf, it can use the RF sensor mounted on its end-effector to identify the cup’s material before making contact and accordingly adjust its grasp [1].
>
> We appreciate the suggestion and have supplemented the application scenarios of the RF-MatID dataset in the Section 1 and 3.2 to refine the scope of our work for better clarification.
>
> **Q2: How does the dataset account for real-world RF conditions beyond geometric variations, such as electromagnetic interference, multipath reflections, environmental variability (humidity), or material variability (surface roughness)?**
>
> **Answer**: Beyond systematically incorporating geometric variations that reflect typical perturbations encountered in indoor embodied AI scenarios, RF-MatID also inherently captures aspects of real-world RF conditions, including material variability and multipath reflections.
>
> * **Multipath effects** are naturally present, as time-domain signals visualized in Appendix Figure 6 and 7 show secondary reflections beyond the direct path.
>
> * **Material variability** (e.g., in density, roughness, and dielectric properties) is inherently encoded in the dataset. For instance, lightweight perforated bricks and lava bricks produce distinct signatures in the raw frequency-domain signals due to materials’ density and surface roughness differences.

---

> ### Author Response · Authors · 2025-11-19
> **Response to Reviewer QnNM (2/5)**
>
> **Q2: How does the dataset account for real-world RF conditions beyond geometric variations, such as electromagnetic interference, multipath reflections, environmental variability (humidity), or material variability (surface roughness)?**
>
> **Answer**: RF-MatID intentionally does not include explicit perturbations from environmental variability or EM interference for the following considerations.
>
> * **Environmental factors**, such as humidity, are relatively controlled in typical indoor embodied AI scenarios and are expected to have a negligible impact on RF signal propagation and material characterization.
>
> * **EM interference** has minimal effect on our radar measurements in indoor settings. The continuous linear frequency modulation of FMCW signals allows for echo separation even under multi-target or overlapping frequency conditions. Moreover, commercial off-the-shelf RF devices overlap with our sensor’s operating bands only in narrow frequency segments (e.g., 0.8 GHz of 5–5.8 GHz WiFi and 1.5 GHz segments of UWB bands), and our sub-band analysis demonstrates that material classification can be reliably performed outside these ranges.
>
> In summary, RF-MatID systematically incorporates geometric variations while inherently capturing material variability and multipath effects, and intentionally excludes environmental or EM interference perturbations because these have minimal impact on indoor UWB-mmWave measurements. Meanwhile, for clarity and precision, we have revised the terminology by replacing ‘perturbation-aware’ and ‘real-world perturbation’ with the more precise terms ‘geometry-diverse’ and ‘geometric perturbation,’ respectively.
>
> **Q3: Could the authors clarify the actual number of unique physical measurements in the dataset and the added value of including both frequency- and time-domain data, given that time-domain samples are computed via inverse FFT and may duplicate information? What new information does the time-domain dataset provide that cannot be derived from the frequency data?**
>
> **Answer**: The RF-MatID contains 71k unique frequency-domain measurements, each paired with a time-domain representation via a standardized IFFT pipeline, resulting in a total of 142k samples across dual domains. While the number of physical measurements reflects the sensing scale, **the representation count better reflects the dataset’s computational richness**. We intentionally include both domains and report the representation-level count because it aligns with established conventions in premier AI benchmarks. Seminal datasets like ImageNet [2] [3] and SUN Database [4] explicitly released and quantified pre-computed feature representations (e.g., SIFT/HOG) alongside raw pixels to facilitate feature-learning research.
>
> The value of providing dual-domain representations could be justified from the following perspectives.
>
> * **Facilitate robust representation learning:** While time-domain data can be derived from frequency-domain signals via IFFT, it provides complementary representational structures that deep networks can exploit. Time-domain inputs emphasize transient and causal structures, whereas frequency-domain representations capture harmonic and spectral regularities. Explicitly providing both domains allows models to leverage this asymmetry directly, rather than implicitly learning the Fourier transform from a single representation, which can otherwise lead to suboptimal optimization and limited generalization. Moreover, paired representations enable studies on how neural networks extract and interpret complementary features from two physically consistent but differently structured views of the same measurement, thereby advancing research on signal representation learning in AI.
>
> * **Enabling Accessible and Reproducible RF Benchmarks:** Providing both domains removes the domain-specific barrier of implementing a standard IFFT, a non-trivial task for AI researchers without a signal processing background. This pre-processing ensures a standardized and unified baseline, facilitating fair model comparison, reproducibility of experiments, and lowering the barrier for broader adoption and benchmarking in AI research.
>
> To address this concern, we have added the clarification in the Introduction line 65 to explicitly distinguish between unique physical measurements and representation-level samples.

---

> ### Author Response · Authors · 2025-11-19
> **Response to Reviewer QnNM (3/5)**
>
> **Q4: How could the benchmark provide insights into the broader applicability of RF sensing for real-world material-based applications beyond material classification?**
>
> **Answer**: Material identification is the cornerstone of downstream material-dependent robotics tasks such as adaptive grasping [1][5], surface-aware interaction [6][7], and affordance-driven policy learning [8][9]. We systematically surveyed existing RF-based material identification datasets, but found that they rely on narrow-band COTS sensors, lack large-scale benchmarks, and are not publicly available, as summarized in Table 1.
>
> Thus, RF-MatID is proposed as the first systematic dataset and benchmark that leverages a compact, wide-band, non–off-the-shelf RF sensor to collect data across fine-grained material categories and diverse geometric perturbations. It provides comprehensive benchmarks with 9 models, 5 evaluation protocols, and 7 data split settings. By validating that deep learning models can achieve fine-grained material identification from RF signals, RF-MatID enables initial material-dependent embodied AI tasks under controlled conditions.
>
> For example, in one of our ongoing robotic manipulation studies, a robotic arm positions an end-effector–mounted RF sensor roughly perpendicular (~90°) to a target object to acquire an ideal RF measurement and predict material category. Based on the identified material, the robot infers the object’s affordance and completes a material-dependent affordance learning task, illustrating the practical applicability of RF-MatID in embodied AI scenarios.
>
> **Q5: Could the authors also evaluate RF-sensing models in the benchmark?**
>
> **Answer**: We appreciate the reviewer’s suggestion. We surveyed existing end-to-end RF-sensing models designed for material identification, selected two representative learning-based methods [10] [11], and evaluated them on RF-MatID to further strengthen the benchmark. The results have been added to Table 2 and are shown below.
>
> |  |  |  | Material-ID [10] |  |  |  |  |  | AirTac [11] |  |
> |:---:|:---:|:---:|:---:|:---:|---|---|---|---|:---:|---|
> |  |  | Protocol 1 | Protocol 2 | Protocol 3 |  |  |  | Protocol 1 | Protocol 2 | Protocol 3 |
> | S1 | - | 96.81% | 93.53% | 97.32% |  |  |  | 99.77% | 91.13% | 98.76% |
> |  | mod1 | 95.67% | 93.20% | 97.55% |  |  |  | 91.36% | 91.16% | 88.84% |
> | S2 | mod2 | 71.59% | 72.43% | 62.83% |  |  |  | 86.95% | 71.91% | 73.84% |
> |  | mod3 | 72.37% | 75.36% | 58.59% |  |  |  | 65.41% | 66.89% | 54.64% |
> |  | mod1 | 97.63% | 86.86% | 98.09% |  |  |  | 98.12% | 88.51% | 95.51% |
> | S3 | mod2 | 51.80% | 53.09% | 82.40% |  |  |  | 76.60% | 69.61% | 87.11% |
> |  | mod3 | 70.54% | 78.49% | 65.75% |  |  |  | 75.23% | 81.45% | 62.77% |
>
> **Q6: Many material samples appear visually and structurally similar, suggesting high intra-class correlation; does the dataset provide sufficient variability in properties such as texture, thickness, or surface roughness to support generalization beyond the collected lab samples?**
>
> **Answer**: RF-MatID has provided sufficient material variability by including multiple sub-class materials within each super-class that differ in thickness, density, surface roughness, and other physical properties. These fine-grained variants were selected to represent common indoor materials in embodied AI tasks.
>
> Furthermore, we intentionally included many visually and structurally similar material samples (e.g., transparent acrylic and tempered glass) in order to demonstrate a key motivation: RF sensing could effectively overcome the inherent limitations that material's visual and structural similarities pose to vision-based approaches [12]. Unlike vision, RF sensing exploits electromagnetic interactions to reveal intrinsic material properties beyond surface appearance. Even structurally similar materials exhibit distinct electromagnetic properties, producing unique signal representations in RF data. For example, variations in permittivity and conductivity affect attenuation (amplitude), while differences in thickness and refractive index alter propagation delay (phase). These physical differences provide the fundamental basis for RF sensing to achieve fine-grained material identification, even when visual or structural cues are insufficient.
>
> To illustrate RF-MatID’s material variability, we compare the properties of sub-class materials within the ‘bricks’ super-class in terms of thickness, density, and surface roughness in the table below. Further detailed descriptions of all material samples are provided in the Appendix Section B.3.
>
> | Material Properties | Thickness (mm) | Density (g/cm^3) | Surface Roughness (μm) |
> |---|---|---|---|
> | (a) Overfired Clay Brick | 60 | 1.786 | 40 |
> | (b) Lightweight Perforated Brick | 30 | 1.667 | 18 |
> | (c) Lava Brick | 25 | 2.160 | 180 |

---

> ### Author Response · Authors · 2025-11-19
> **Response to Reviewer QnNM (4/5)**
>
> **Q7: The use of the term “mod” for both modality and mode is confusing and should be clarified in the benchmark descriptions to avoid misinterpretation.**
>
> **Answer**: We appreciate the reviewer’s suggestion and have added an explanatory sentence of the use of the term “mod” in the section 5.1 line 413 for better clarification. The term “mod” we used in the benchmark tables refers to the data split sub-modes defined in the section 4.2 “Data Splits” paragraph.
>
> **Q8: Are the perturbations (distance and angle) annotated per sample, and are these metadata available in the dataset?**
>
> **Answer**: Yes, both geometric perturbations are systematically incorporated and explicitly annotated for each sample, as illustrated in the dataset batch provided in the Supplementary Material. In RF-MatID, representations in each domain are organized into folders according to material–distance–angle combinations. Folder names follow the format: “ms46131a-4To43.5GHz-2048-<material_category>-<distance>-<angle>” (e.g., the folder “ms46131a-4To43.5GHz-2048-brick01-200mm-0deg” contains overfired clay brick samples collected at a 200 mm placement distance and 0° incidence angle), providing explicit metadata to support flexible dataset usage.
>
> **Q9: Are there more textual annotations that we could use to give LLM the capabilities to describe the material?**
>
> **Answer**: Yes, detailed textual descriptions for each material category are provided in the Appendix section B.3. These descriptions include physical properties such as dimensions, density, and surface texture (e.g., “Overfired Clay Brick: rectangular, 210 × 100 × 60 mm, approximate density 1.786 g/cm³, hard vitrified surface, low porosity”), as well as intended usage and application scenarios (e.g. “suitable for high-strength masonry, paving, or other applications where durability and a rustic, speckled appearance are desired”). Such rich textual annotations provide semantic information that is particularly relevant for affordance learning in embodied AI and enable LLM to interpret and reason about material properties.
>
> **Q10: The paper reports high accuracy (>99%) across models, could this indicate potential overfitting due to low intra-class diversity or lack of environmental variability? Is the task sufficiently challenging to drive future methodological improvement?**
>
> **Answer**: As shown in Table 2, **models achieve >99% accuracy only under the data-split configuration of Setting 1, i.e., the ideal condition where the dataset is uniformly splitted, whereas accuracy drops below 50% under more challenging cross-distance settings.** Hence, our dataset has been very challenging for future methodological improvement.
>
> In RF-MatID, task difficulty is jointly determined by the 5 frequency-band protocols and 7 data split settings. Narrower spectral bands (e.g., P4/P5 follow legally permitted bands in major economies) reduce physical discriminability, while split settings introduce systematic distribution shifts (e.g., S2 varies sensing distance), substantially challenging models’ ability to generalize across geometric-domain variations.
>
> These challenges reflect realistic conditions in indoor embodied AI. For instance, in a robotic manipulation task (e.g., retrieving a glass cup from a shelf), the end-effector's RF sensor must capture signals under varying hand–object distances, changing incidence angles, and bandwidth constraints. This allows the robot to seamlessly integrate pre-contact material identification into the manipulation process, thereby accordingly adjusting its grasping policy (such as adjusting force or contact points) [13]. Under such complex configurations, model performance drops substantially (e.g., from >99% in S1 to 72% in S2 and 86% in S3).
>
> In future work, we plan to expand the material categories, incorporate finer-grained material variability, and deploy the RF sensor on real robotic platforms for in-situ data collection. As the task grows in realism and complexity, it will become substantially more challenging and remains far from being solved for real-world deployment.

---

> ### Author Response · Authors · 2025-11-19
> **Response to Reviewer QnNM (5/5)**
>
> **Reference:**
>
> [1] Sundermeyer, M., Mousavian, A., Triebel, R., & Fox, D. (2021, May). Contact-graspnet: Efficient 6-dof grasp generation in cluttered scenes. In 2021 IEEE International Conference on Robotics and Automation (ICRA) (pp. 13438-13444). IEEE.
>
> [2] Deng, J., Dong, W., Socher, R., Li, L. J., Li, K., & Fei-Fei, L. (2009, June). Imagenet: A large-scale hierarchical image database. In 2009 IEEE conference on computer vision and pattern recognition (pp. 248-255). Ieee.
>
> [3] Russakovsky, O., Deng, J., Su, H., Krause, J., Satheesh, S., Ma, S., ... & Fei-Fei, L. (2015). Imagenet large scale visual recognition challenge. International journal of computer vision, 115(3), 211-252.
>
> [4] Xiao, J., Hays, J., Ehinger, K. A., Oliva, A., & Torralba, A. (2010, June). Sun database: Large-scale scene recognition from abbey to zoo. In 2010 IEEE computer society conference on computer vision and pattern recognition (pp. 3485-3492). IEEE.
>
> [5] Mahler, J., Matl, M., Satish, V., Danielczuk, M., DeRose, B., McKinley, S., & Goldberg, K. (2019). Learning ambidextrous robot grasping policies. Science Robotics, 4(26), eaau4984.
>
> [6] Wang, S., She, Y., Romero, B., & Adelson, E. (2021, May). Gelsight wedge: Measuring high-resolution 3d contact geometry with a compact robot finger. In 2021 IEEE International Conference on Robotics and Automation (ICRA) (pp. 6468-6475). IEEE.
>
> [7] Yang, F., Ma, C., Zhang, J., Zhu, J., Yuan, W., & Owens, A. (2022). Touch and go: Learning from human-collected vision and touch. arXiv preprint arXiv:2211.12498.
>
> [8] Mo, K., Qin, Y., Xiang, F., Su, H., & Guibas, L. (2022, January). O2o-afford: Annotation-free large-scale object-object affordance learning. In Conference on robot learning (pp. 1666-1677). PMLR.
>
> [9] Mo, K., Guibas, L. J., Mukadam, M., Gupta, A., & Tulsiani, S. (2021). Where2act: From pixels to actions for articulated 3d objects. In Proceedings of the IEEE/CVF International Conference on Computer Vision (pp. 6813-6823).
>
> [10] Chen, G., Luo, C., Zeng, H., Wen, G., Luo, Z., Wang, J., ... & Li, J. (2025). Material-ID: Towards mmWave-based Material Identification. ACM Transactions on Sensor Networks, 21(4), 1-26.
>
> [11] Zhang, Z., Song, D., Zhou, A., & Ma, H. (2024). airTac: A Contactless Digital Tactile Receptor for Detecting Material and Roughness via Terahertz Sensing. Proceedings of the ACM on Interactive, Mobile, Wearable and Ubiquitous Technologies, 8(3), 1-37.
>
> [12] Khushaba, R. N., & Hill, A. J. (2022). Radar-based materials classification using deep wavelet scattering transform: A comparison of centimeter vs. millimeter wave units. IEEE Robotics and Automation Letters, 7(2), 2016-2022.
>
> [13] Murali, A., Mousavian, A., Eppner, C., Paxton, C., & Fox, D. (2020, May). 6-dof grasping for target-driven object manipulation in clutter. In 2020 IEEE International Conference on Robotics and Automation (ICRA) (pp. 6232-6238). IEEE.

---

> ### Comment · Reviewer_QnNM · 2025-11-20
> **Response to authors' rebuttal**
>
> Dear Authors,
>
> Thank you very much for your detailed responses, which have addressed most of my concerns. Given the novelty of the work, the revision with corrected terms, and the comprehensiveness of the evaluation, I am willing to raise my scores accordingly.
>
> Good luck with the submission!

---

> > ### Author Response · Authors · 2025-11-22
> > **Appreciation for the constructive comments**
> >
> > We sincerely appreciate your helpful suggestions! Thanks for your time and effort!

---

### Official Review · Reviewer_FgEZ · 2025-10-31

**Soundness:** 3
**Presentation:** 3
**Contribution:** 2
**Rating:** 4
**Confidence:** 3

**Summary:**

This paper introduces a large-scale dataset and benchmark (RF-MatID) for radio-frequency–based material identification.
The dataset spans 16 material types across 5 superclasses.
It includes realistic perturbations such as varying angles and distances to test robustness.
The authors further establish multiple benchmark protocols to evaluate state-of-the-art models under both standard and distribution-shift conditions.
The work aims to standardize evaluation and accelerate progress in RF-based material recognition for real-world applications.

**Strengths:**

- This work addresses a current gap in RF-based material sensing by providing a new dataset and benchmark focused on material identification.
- The dataset is relatively large, covers a wide frequency range, and incorporates variations in acquisition conditions.
- The dataset includes of both time- and frequency-domain representations, along with several evaluation protocols targeting both in-distribution performance and robustness to angle and distance shifts.
- The authors also benchmark several state-of-the-art deep learning models across these protocols, establishing initial baselines for comparison and facilitating future reproducible work in this area.

**Weaknesses:**

- More discussion on the dataset's limitations, potential biases, and cost or practicality of data collection would help contextualize the scope and applicability of the benchmark.
- While the dataset is extensive, the paper would benefit from further clarification on the real-world representativeness of the collection setup and whether the acquisition hardware and environments generalize beyond the authors' configuration.
- The evaluation focuses primarily on standard deep learning baselines, and it is unclear how classical RF signal-processing methods or hybrid approaches would perform under the same protocols.
- Although perturbations in angle and distance are included, other sources of real-world variability are not explored in depth.

**Questions:**

See weaknesses

---

> ### Author Response · Authors · 2025-11-19
> **Response to Reviewer FgEZ (1/3)**
>
> We appreciate the Reviewer FgEZ for the insightful questions and constructive feedback. We are glad that the reviewer acknowledges our work **“addresses a current gap in RF-based material sensing”** and provides valuable suggestions. To improve our manuscript, we have worked hard to address the lack of discussions on the dataset's limitations, potential biases, cost or practicality of data collection, and real-world representativeness through providing additional physics-level discussions and elaborations in the manuscript to enhance RF-MatID’s contributions and impacts towards realistic embodied AI applications. Moreover, we have added experiments on RF-based material identification methods to provide a more comprehensive and informative benchmark.
>
> **Q1: To better contextualize the benchmark’s scope and applicability, more discussion could be provided on the dataset’s limitations, potential biases, and the cost or practicality of data collection.**
>
> **Answer**: We appreciate the suggestion and realize the importance of contextualizing the benchmark’s scope and applicability. We have added more discussion to elaborate on the dataset’s limitations and potential biases both in Section 6 and below, which also highlights directions for our future work.
>
> 1.    Controlled acquisition setup: data is collected in a controlled platform rather than on deployed robots, potentially missing operational biases. Real-world robotic deployments will be involved in future releases.
>
> 2.    Material diversity: the dataset covers 16 fine-grained indoor materials but could include richer material categories, multilayer material variations, and variations with fine-grained properties in the future.
>
> 3.    Spectrum width: The frequency band presently spans 4–43.5 GHz and could extend to broader and higher-frequency bands in future datasets.
>
> 4.    Environmental variability: instead of current lab environments, more diverse application environments such as within a kitchen or a rehabilitation room could be incorporated to better evaluate robustness.
>
> Additional demonstrations are also provided in Section 3.2 and 4.3 to reveal the applicability of the proposed data collection setup in realistic indoor embodied AI scenarios. To support this, RF-MatID employs a compact, customized single-input–single-output (SISO) FMCW radar designed specifically for indoor embodied AI. The sensor is lightweight and low-cost, making it practical to mount on a robot end-effector and readily integrable into manipulation or affordance-learning pipelines. Compared with commercial UWB/mmWave devices [1] [2], the proposed sensor also offers a smaller beam footprint (i.e. the width of the beam on the material surface), enabling fine-grained, local material characterization that aligns with the requirements of indoor embodied perception, including for small-sized objects.
>
> **Q2: How representative the data collection setup is of real-world scenarios and whether the acquisition hardware and environments can generalize beyond the authors’ specific configuration?**
>
> **Answer**: This hardware and data collection setup was motivated by our goal of designing a new RF sensor in a manner analogous to designing a RGB camera in computational photography. Just as computational photography optimizes camera hardware and acquisition pipelines to capture richer visual information [3], we approached the problem from a computational RF perspective, aiming to identify an optimal hardware and data collection configuration that yields high-quality RF signals suitable for material identification. Only after establishing this foundation can the setup be extended to more realistic embodied AI applications.
>
> We explored commercial off-the-shelf RF sensors [1] [2] but found that they cannot meet the task requirements or support future studies due to fragmented frequency coverage, insufficient informational richness, and large beam footprints, which have been customized for specific applications of off-the-shelf RF sensors. This motivated the development of the non–off-the-shelf RF sensor and the custom data collection setup specifically designed for indoor embodied AI scenarios presented in this work.
>
> The RF-MatID setup represents typical indoor embodied AI scenarios by including fine-grained, common indoor materials and systematically incorporating geometric perturbations, such as varying sensor–object distances and incidence angles, that reflect realistic sensing conditions for sensors mounted on manipulators [4].
>
> The customized hardware is broadly generalizable to real-world applications. The compact single-input–single-output (SISO) frequency-modulated continuous wave (FMCW) radar can be mounted on various robot end-effectors or mobile platforms to capture material characteristics across diverse indoor layouts. Its small beam footprint enables accurate classification of individual objects, including small items like teacups, under typical placement and layout variations.

---

> ### Author Response · Authors · 2025-11-19
> **Response to Reviewer FgEZ (2/3)**
>
> **Q3: How classical RF signal-processing methods or hybrid approaches would perform under the same evaluation protocols, given that the current evaluation focuses primarily on standard deep learning baselines?**
>
> **Answer**: We appreciate the reviewer’s suggestion. We have surveyed classical RF signal-processing and hybrid approaches specifically proposed for material identification, selected two representative methods, mSense [5] and RFVibe [6], and evaluated them under the same RF-MatID protocols. The results have been incorporated and discussed in Appendix Section C.5 and are shown below.
>
> We observe that mSense fails to distinguish the fine-grained material categories, primarily because classical methods rely on background-only measurements for noise removal, whereas our dataset trains directly on signals containing background noise. This highlights the advantage and necessity of learning-based models for RF material identification.
>
> Signal-processing method:
>
> | | |  | mSense [5] |  |
> |:---:|:---:|:---:|:---:|:---:|
> |  |  | Protocol 1 | Protocol 2 | Protocol 3 |
> | S1 | - | 10.31% | 8.53% | 10.05% |
> |  | mod1 | 9.91% | 8.47% | 10.33% |
> | S2 | mod2 | 13.10% | 10.25% | 10.74% |
> |  | mod3 | 8.51% | 7.95% | 8.20% |
> |  | mod1 | 10.38% | 9.74% | 9.09% |
> | S3 | mod2 | 7.50% | 6.38% | 10.30% |
> |  | mod3 | 6.42% | 6.81% | 6.42% |
>
> Two-stage hybrid method:
>
> |  | | | RFVibe [6] |  |
> |:---:|:---:|:---:|:---:|:---:|
> |  |  | Protocol 1 | Protocol 2 | Protocol 3 |
> | S1 | - | 83.76% | 86.48% | 79.06% |
> |  | mod1 | 80.48% | 82.51% | 91.09% |
> | S2 | mod2 | 50.36% | 52.16% | 50.28% |
> |  | mod3 | 72.20% | 82.70% | 67.93% |
> |  | mod1 | 83.45% | 86.50% | 80.27% |
> | S3 | mod2 | 55.97% | 69.59% | 75.60% |
> |  | mod3 | 76.48% | 92.93% | 71.28% |
>
> **Q4: How other sources of real-world variability, beyond angle and distance perturbations, are explored in the dataset?**
>
> **Answer**: Beyond systematically incorporating geometric variations that reflect typical perturbations encountered in indoor embodied AI scenarios, RF-MatID also inherently captures aspects of real-world RF conditions, including material variability and multipath reflections.
>
> * **Material variability** (e.g., in density, roughness, and dielectric properties) is inherently encoded in the dataset. For instance, lightweight perforated bricks and lava bricks produce distinct signatures in the raw frequency-domain signals due to materials’ density and surface roughness differences.
>
> * **Multipath effects** are also naturally present, as time-domain signals visualized in Appendix Figure 5 showing secondary reflections beyond the direct path.
>
> RF-MatID intentionally excludes explicit perturbations from environmental variability, mechanical vibrations, or electromagnetic (EM) interference based on the following considerations.
>
> * **Environmental factors**, such as humidity, are relatively controlled in typical indoor embodied AI scenarios and are expected to have a negligible impact on RF signal propagation and material characterization.
>
> * **Mechanical vibrations** are typically compensated by the robot’s control algorithms.
>
> * **EM interference** has minimal effect on our FMCW radar measurements in indoor settings. The continuous linear frequency modulation of FMCW signals allows echo separation even under multi-target or overlapping frequency conditions . Moreover, commercial off-the-shelf RF devices overlap with our sensor’s operating bands only in narrow frequency segments (e.g., 0.8 GHz of 5–5.8 GHz WiFi and 1.5 GHz segments of UWB bands), and our sub-band analysis demonstrates that material classification can be reliably performed outside these ranges.
>
> We have provided a discussion of additional forms of realistic perturbations beyond geometric variations in Appendix B.4

---

> ### Author Response · Authors · 2025-11-19
> **Response to Reviewer FgEZ (3/3)**
>
> **Reference:**
>
> [1] Novelda AS. (2017). X4 datasheet: Ultra-wideband (UWB) impulse radar transceiver SoC (Rev. E). [Datasheet]. Retrieved from https://novelda.com/technology/datasheets/
>
> [2] Texas Instruments. (2019). IWR6843ISK-ODS mmWave sensor evaluation module user’s guide (SPRUIS6). [User's Guide]. Retrieved from https://www.ti.com/tool/IWR6843ISK-ODS
>
> [3] Sitzmann, V., Diamond, S., Peng, Y., Dun, X., Boyd, S., Heidrich, W., ... & Wetzstein, G. (2018). End-to-end optimization of optics and image processing for achromatic extended depth of field and super-resolution imaging. ACM Transactions on Graphics (TOG), 37(4), 1-13.
>
> [4] Mahler, J., Matl, M., Satish, V., Danielczuk, M., DeRose, B., McKinley, S., & Goldberg, K. (2019). Learning ambidextrous robot grasping policies. Science Robotics, 4(26), eaau4984.
>
> [5] Wu, C., Zhang, F., Wang, B., & Liu, K. R. (2020). mSense: Towards mobile material sensing with a single millimeter-wave radio. Proceedings of the ACM on Interactive, Mobile, Wearable and Ubiquitous Technologies, 4(3), 1-20.
>
> [6] Shanbhag, H., Madani, S., Isanaka, A., Nair, D., Gupta, S., & Hassanieh, H. (2023, June). Contactless material identification with millimeter wave vibrometry. In Proceedings of the 21st Annual International Conference on Mobile Systems, Applications and Services (pp. 475-488).

---

> ### Author Response · Authors · 2025-11-27
> **Looking forward to your reply**
>
> Dear Reviewer FgEZ,
>
> As the rebuttal deadline is approaching, could you please have a look at our response? We are looking forward to your reply!
>
> Feel free to let us know if you have any other concerns. Thanks for your time and effort!
>
> Best Regards,
>
> Authors of Submission 9536

---

### Official Review · Reviewer_vQa1 · 2025-11-01

**Soundness:** 4
**Presentation:** 3
**Contribution:** 2
**Rating:** 6
**Confidence:** 3

**Summary:**

Focusing on material identification, this work indicates that there lacks a large-scale datasets with real materials labeled by frequency signals. According to this paper, frequency-domain signal data can not only increase the material identification, but also enhance the generalizability to real-world applications where a domain gap always exists. Therefore, this work proposes a frequency-labeled dataset with 142k samples, 16 fine-grained categories.

Moreover, this work integrated a benchmark that evaluates 8 previous works and one proposed baseline. The benchmark tests the domain adaptation ability, reveals some existing challenges.

**Strengths:**

1. Large dataset and effective data preprocessing: This work collected 71k frequency samples and extend them to time-domain representation, which is a good contribution for real-world applications. In addition, the data preprocessing is carefully designed to augment the data sources.
2. Comprehensive benchmark: The benchmark evaluated different protocols for different real-world usage. The domain adaptation evaluation is conducted on reasonable data split strategies, and the hierachical label split can effectively reveal the model's capability for material identification.
3. This work is well motivated and has the potential to inspire domain experts to design and test new framework for material identification.

**Weaknesses:**

1. Presentation: The presentation involves too many mechanical material-specific details, while, correspondingly, lacks AI-oriented intuitions, which is not friendly for a broader community. For example, how the details of Eq. (1) contribute to this design.
2. The benchmarked methods are all general methods that originally proposed for other downstream tasks. Is there any material identification specific methods, or frequency process focused methods worth to be includded?
3. The contributions to the AI community still remain to be discussed. The authors mentioned that the dataset can contribute to multi-modal learning. However, this is a weak claim without further demonstrations.

**Questions:**

Please address the questions in weaknesses if possible.

---

> ### Author Response · Authors · 2025-11-19
> **Response to Reviewer vQa1 (1/2)**
>
> We are deeply grateful to Reviewer vQa1 for the remarkably perceptive AI-centered suggestions and the outstanding level of domain expertise demonstrated in the review. Reviewer’s thorough reading, constructive insights, and forward-looking perspectives have significantly strengthened the clarity, depth, and technical rigor of our work.  We sincerely appreciate the reviewer’s recognition that RF-MatID **“is well motivated”**, contributing **“large dataset and effective data preprocessing”** and **“comprehensive benchmark”**. Based on the suggestions, we have addressed the lack of AI-oriented intuitions and demonstrations by providing AI-inspired discussions and elaborations in the manuscript to enhance RF-MatID’s contributions and impacts towards the AI community . In addition, we have added experiments on RF-based material identification approaches to provide a more comprehensive benchmark.
>
> **Q1: Could authors provide more AI-oriented intuitions along with the presentation of mechanical material-specific details? For example, how the details of Eq. (1) contribute to this design.**
>
> **Answer**: We appreciate the reviewer’s suggestion and have added more AI-oriented intuitions alongside the mechanical material-specific details in Sections 3 and 4, providing a presentation that is more accessible to the AI community.
>
> Equation (1) defines the raw complex-valued measurements, which physically encode amplitude attenuation (magnitude) and propagation delay (phase). Since standard deep learning backbones operate in the real domain, this presents a critical architectural choice: whether to employ specialized Complex-Valued Neural Networks (CVNNs) [1] or to project the data into a dual-channel real-valued representation. Our experiments in Section 4.2 demonstrate that the dual-channel approach is superior. It allows standard models to effectively learn the latent interactions between amplitude and phase, yielding improved out-of-distribution (OOD) generalization and more accurate fine-grained classification.
>
> Apart from model-friendly representation design, we believe the following AI-oriented intuitions based on mechanical material-specific details are also worth discussing and have been added in Appendix Section B.5:
>
> 1.    Embedded Physical Constraints: In physical science, radar equations and other physical laws are well studied. Incorporate them as regularization terms or hard constraints in the model could guide feature learning according to known material electromagnetic responses [2].
>
> 2.    Disentangled Representation Learning: In material science, intrinsic material properties (e.g., permittivity, density) can be separated from geometric factors (e.g., distance, incidence angle). Incorporating disentangled representation learning [3] could guide the model to capture geometry-invariant material features while representing geometric variations linearly.
>
> 3.    Spectral Attention: In RF sensing, materials are identified by unique characteristics occurring at certain frequencies (e.g., periodic fluctuations or sharp energy changes) that reflect their thickness and internal structure rather than overall signal strength. Frequency-domain attention [4] can guide models to focus on the most informative frequency spectrum for material discrimination.
>
> We sincerely appreciate the expert-level suggestions. The AI-oriented insights have greatly sharpened the manuscript’s clarity and meaningfully strengthened the broader impact of our work on the AI community.
>
> **Q2: Are there any material identification-specific methods or frequency-processing-focused methods that should be included in the benchmark, beyond the general methods originally proposed for other downstream tasks?**
>
> **Answer**: Yes, we have surveyed classical RF signal-processing approaches, hybrid methods, and end-to-end RF-sensing models specifically designed for material identification, respectively selected four representative methods mSense [5], RFVibe [6], Material-ID [7] & airTac [8], and have evaluated them under the same RF-MatID protocols. The results have been incorporated into Table 2 and Appendix C.5, and are also shown below.
>
> We observe that mSense fails to distinguish the fine-grained material categories, primarily because classical methods rely on background-only measurements for noise removal, whereas our dataset trains directly on signals containing background noise. This highlights the advantage and necessity of learning-based models for RF material identification.
>
> Signal-processing method:
>
> | | |  | mSense [5] |  |
> |:---:|:---:|:---:|:---:|:---:|
> |  |  | Protocol 1 | Protocol 2 | Protocol 3 |
> | S1 | - | 10.31% | 8.53% | 10.05% |
> |  | mod1 | 9.91% | 8.47% | 10.33% |
> | S2 | mod2 | 13.10% | 10.25% | 10.74% |
> |  | mod3 | 8.51% | 7.95% | 8.20% |
> |  | mod1 | 10.38% | 9.74% | 9.09% |
> | S3 | mod2 | 7.50% | 6.38% | 10.30% |
> |  | mod3 | 6.42% | 6.81% | 6.42% |

---

> ### Author Response · Authors · 2025-11-19
> **Response to Reviewer vQa1 (2/2)**
>
> **Q2: Are there any material identification-specific methods or frequency-processing-focused methods that should be included in the benchmark, beyond the general methods originally proposed for other downstream tasks?**
>
> **Answer**:
>
> Two-stage hybrid method:
>
> |  | | | RFVibe [6] |  |
> |:---:|:---:|:---:|:---:|:---:|
> |  |  | Protocol 1 | Protocol 2 | Protocol 3 |
> | S1 | - | 83.76% | 86.48% | 79.06% |
> |  | mod1 | 80.48% | 82.51% | 91.09% |
> | S2 | mod2 | 50.36% | 52.16% | 50.28% |
> |  | mod3 | 72.20% | 82.70% | 67.93% |
> |  | mod1 | 83.45% | 86.50% | 80.27% |
> | S3 | mod2 | 55.97% | 69.59% | 75.60% |
> |  | mod3 | 76.48% | 92.93% | 71.28% |
>
> End-to-end RF-sensing methods:
>
> |  |  |  | Material-ID [7] |  |  |  |  |  | AirTac [8] |  |
> |:---:|:---:|:---:|:---:|:---:|---|---|---|---|:---:|---|
> |  |  | Protocol 1 | Protocol 2 | Protocol 3 |  |  |  | Protocol 1 | Protocol 2 | Protocol 3 |
> | S1 | - | 96.81% | 93.53% | 97.32% |  |  |  | 99.77% | 91.13% | 98.76% |
> |  | mod1 | 95.67% | 93.20% | 97.55% |  |  |  | 91.36% | 91.16% | 88.84% |
> | S2 | mod2 | 71.59% | 72.43% | 62.83% |  |  |  | 86.95% | 71.91% | 73.84% |
> |  | mod3 | 72.37% | 75.36% | 58.59% |  |  |  | 65.41% | 66.89% | 54.64% |
> |  | mod1 | 97.63% | 86.86% | 98.09% |  |  |  | 98.12% | 88.51% | 95.51% |
> | S3 | mod2 | 51.80% | 53.09% | 82.40% |  |  |  | 76.60% | 69.61% | 87.11% |
> |  | mod3 | 70.54% | 78.49% | 65.75% |  |  |  | 75.23% | 81.45% | 62.77% |
>
> **Q3: How would RF-MatID contribute to the AI community? More demonstrations (especially about the contribution to multi-modal learning) could be included to strengthen the contributions.**
>
> **Answer**: We appreciate the reviewer’s contributive suggestion. In addition to the dataset and benchmark contributions described in the manuscript, we highlight RF-MatID’s broader value to the AI community from both the modality and hardware perspectives.
>
> **From a modality perspective,** RF-MatID offers an RF sensing modality tailored to capture materials’ electromagnetic properties. The RF modality complements other sensing modality, such as vision or acoustic, and enables multimodal learning frameworks to jointly reason over spatial structure, instruction semantics, and material properties. For example, fusing RF-derived local material characteristics with visual features can enhance indoor scene understanding, helping embodied agents infer object affordances, spatial layouts, and physical interactions in cluttered indoor environments.
>
> **From a hardware perspective,** RF-MatID introduces a compact, lightweight UWB–mmWave platform suitable for mounting on a robot end-effector to perform fine-grained, real-time material characterization. This supports the embodied AI community to develop material-aware manipulation and affordance-driven workflows, such as selecting grasp strategies based on material compliance, adjusting contact forces according to surface hardness, and predicting material-grounded affordances (e.g., graspable, cuttable, pourable).
> We have expanded Section 4.3 to provide concrete demonstrations of RF-MatID’s contributions to the AI community.
>
> **Reference:**
>
> [1] Hirose, A. (2006). Complex-valued neural networks. Berlin, Heidelberg: Springer Berlin Heidelberg.
>
> [2] Raissi, M., Perdikaris, P., & Karniadakis, G. E. (2019). Physics-informed neural networks: A deep learning framework for solving forward and inverse problems involving nonlinear partial differential equations. Journal of Computational physics, 378, 686-707.
>
> [3] Higgins, I., Matthey, L., Pal, A., Burgess, C., Glorot, X., Botvinick, M., ... & Lerchner, A. (2017, February). beta-vae: Learning basic visual concepts with a constrained variational framework. In International conference on learning representations.
>
> [4] Hu, J., Shen, L., & Sun, G. (2018). Squeeze-and-excitation networks. In Proceedings of the IEEE conference on computer vision and pattern recognition (pp. 7132-7141).
>
> [5] Wu, C., Zhang, F., Wang, B., & Liu, K. R. (2020). mSense: Towards mobile material sensing with a single millimeter-wave radio. Proceedings of the ACM on Interactive, Mobile, Wearable and Ubiquitous Technologies, 4(3), 1-20.
>
> [6] Shanbhag, H., Madani, S., Isanaka, A., Nair, D., Gupta, S., & Hassanieh, H. (2023, June). Contactless material identification with millimeter wave vibrometry. In Proceedings of the 21st Annual International Conference on Mobile Systems, Applications and Services (pp. 475-488).
>
> [7] Chen, G., Luo, C., Zeng, H., Wen, G., Luo, Z., Wang, J., ... & Li, J. (2025). Material-ID: Towards mmWave-based Material Identification. ACM Transactions on Sensor Networks, 21(4), 1-26.
>
> [8] Zhang, Z., Song, D., Zhou, A., & Ma, H. (2024). airTac: A Contactless Digital Tactile Receptor for Detecting Material and Roughness via Terahertz Sensing. Proceedings of the ACM on Interactive, Mobile, Wearable and Ubiquitous Technologies, 8(3), 1-37.

---

> ### Author Response · Authors · 2025-11-27
> **Looking forward to your reply**
>
> Dear Reviewer vQa1,
>
> As the rebuttal deadline is approaching, could you please have a look at our response? We are looking forward to your reply!
>
> Feel free to let us know if you have any other concerns. Thanks for your time and effort!
>
> Best Regards,
>
> Authors of Submission 9536

---

### Official Review · Reviewer_VGTy · 2025-11-01

**Soundness:** 3
**Presentation:** 3
**Contribution:** 3
**Rating:** 6
**Confidence:** 3

**Summary:**

This paper presents RF-MatID, a large open-source dataset for radio-frequency material identification. It contains 142,000 samples across 16 categories and 5 superclasses, covering 4–43.5 GHz in both frequency and time domains, with controlled variations in angle and distance. The authors benchmark nine deep learning models under multiple frequency and data split settings, showing that raw frequency-domain data can be effectively used. The main contributions include the creation of this dataset, comprehensive benchmarking protocols, and analysis of model robustness and frequency-band effectiveness.

**Strengths:**

1. First large-scale and open-source RF dataset covering a wide frequency range. The scale and diversity of this dataset can well benifit the community of RF-related machine learning-based study.
2. This paper establishes a well-structured benchmark with multiple frequency protocols, data splits, and nine deep learning models, offering a comprehensive evaluation of model robustness.
3. The authors carefully analyzes both frequency and time domain representations. This leads to the insight that frequency-domain data alone can achieve high accuracy, which simplifies model design.
4. The dataset is well organized, rather than just collected. This facilitates future research in RF sensing and cross-domain generalization.

**Weaknesses:**

1. The term "perturbation-aware" is slightly overclaimed. The authors introduce "perturbation" by varying the distance and angle, which is controllable as part of the measuring/sensing technique itself. As the authors mention, the "real-world interference" is not well considered, like electromagnetic noise, mechanical vibrations, etc.
2. One minor suggestion is that some acronyms should be explained before used, like UWB and MMW. Besides, I am not sure whether "mmWave" and MMW mean the same concept. If so, it seems inconsistent.

**Questions:**

1. The authors claim that "the system achieves ∼10 cm spatial resolution". The concept of "spatial resolution"is not very clear to me.
2. I notice in Figure 2 that different material plates have different sizes. Does this setting introduces additional bias into the RF data for each class? For example, a larger plate will make the signal different in some ways?

---

> ### Author Response · Authors · 2025-11-19
> **Response to Reviewer VGTy (1/2)**
>
> We sincerely appreciate Reviewer VGTy’s expert-level suggestions, meticulous reading of the manuscript, detailed review questions, and domain-specific insights, all of which have substantially improved the clarity, conceptual depth, and technical rigor of our work. We greatly appreciate the reviewer’s recognition that our work proposes **“first large-scale and open-source RF dataset covering a wide frequency range”** and **“establishes a well-structured benchmark”**. Guided by the suggestions, we have carefully revised issues related to terminology precision, perturbation definitions, acronym consistency, and the clarity of sensing concepts. Corresponding updates have been made to Sections 3.2 and 4.2, as well as Appendix B and D, with all modifications clearly highlighted in magenta.
>
> **Q1: How does the proposed “perturbation-aware” dataset genuinely reflect real-world perturbations, given that the current variations (distance, angle) are controllable sensing parameters rather than true environmental interferences (e.g., electromagnetic noise, mechanical vibrations)?**
>
> **Answer**: We very much appreciate the reviewer’s in-depth comment and suggestion regarding the term “perturbation-aware” and realize that it does not accurately reflect our dataset scope.
>
> RF-MatID is primarily designed for indoor embodied AI applications like robotic manipulation and affordance learning—for example, a robot using a compact RF sensor to learn and perform adaptive grasping based on material properties. **In such scenarios, RF-MatID reflects realistic perturbations through systematically incorporating controlled geometric variations and inherently capturing material variability and multipath reflections.**
>
> The two geometric variations, **distance and angle**, are widely recognized in RF sensing literature as the primary factors influencing the measured signal characteristics [1][2]. They also mirror realistic conditions in indoor embodied AI applications, where compact sensors mounted on manipulators must recognize materials under varying hand–object distances and changing incidence angles during motion. In our dataset, both variables are explicitly annotated for each sample and released as metadata, facilitating future research on systematically studying and mitigating their impact on model robustness and generalization.
>
> Beyond geometric variations, RF-MatID also inherently captures aspects of real-world RF conditions, including material variability and multipath reflections.
>
> * **Material variability** (e.g., in density, roughness, and dielectric properties) is inherently reflected in the dataset. For instance, lightweight perforated bricks and lava bricks produce distinct signatures in the raw frequency-domain signals due to their density and surface roughness differences.
>
> * **Multipath effects** are also naturally present, as time-domain signals visualized in Appendix Figure 6 & 7 show multiple reflections beyond the direct path.
>
> RF-MatID intentionally does not include explicit perturbations from electromagnetic (EM) noise and mechanical vibrations based on the following considerations.
>
> * **Mechanical vibrations** are typically compensated by the control algorithms [3] [4] of embodied agents operating in indoor scenarios.
>
> * **EM interference** has minimal effect on our radar measurements in indoor settings. The continuous linear frequency modulation of FMCW signals allows echo separation even under multi-target or overlapping frequency conditions . Moreover, commercial off-the-shelf RF devices overlap with our sensor’s operating bands only in narrow frequency segments (e.g., 0.8 GHz of 5–5.8 GHz WiFi and 1.5 GHz segments of UWB bands), and our sub-band analysis demonstrates that material classification can be reliably performed outside these ranges.
>
> In summary, RF-MatID systematically incorporates geometric variations while inherently capturing material variability and multipath effects, and intentionally excludes EM interference and mechanical vibrations perturbations because they have minimal impact on indoor UWB-mmWave measurements.
>
> For clarity and precision, we have revised all the “perturbation-aware” and “real-world perturbation” terms to “geometry-diverse” and “geometric perturbation” in the paper, respectively. We also have added descriptions in Section 4.2 to explain the motivation for incorporating geometric variations that reflect realistic conditions in indoor embodied AI scenarios. We have further provided a discussion of additional forms of realistic perturbations beyond geometric variations in Appendix B.4
>
> We sincerely appreciate the expert-level suggestions. The domain-specific insights have substantially enhanced the manuscript’s clarity and overall scholarly rigor.

---

> ### Author Response · Authors · 2025-11-19
> **Response to Reviewer VGTy (2/2)**
>
> **Q2: Could the authors clarify the usage and consistency of acronyms such as UWB and MMW/mmWave? Are “mmWave” and “MMW” intended to refer to the same sensing modality?**
>
> **Answer**: We appreciate the reviewer’s suggestion and have added explanations for the acronyms at their first occurrence (line 148, 155, 160). Specifically, “UWB” stands for Ultra-WideBand, and “mmWave” stands for millimeter wave. We would also like to clarify that “mmWave” and “MMW” refer to the same sensing modality, with the difference arising from community-specific conventions [1] [5]. To ensure consistency, we have standardized the terminology to “mmWave” throughout the paper.
>
> **Q3: What is the clear definition of  “spatial resolution” in the statement that the system achieves approximately 10 cm spatial resolution?**
>
> **Answer**: In our sensing system, spatial resolution is equivalent to the beam footprint, defining the width of the beam on the material surface. Unlike the notion of spatial resolution in computer vision, in RF sensing a smaller spatial resolution (beam footprint) indicates a more focused beam with higher energy concentration, allowing clearer measurement of local material characteristics. The beam footprint increases with stand-off distance, and thus, in our setup, we added a lens to focus the beam.
>
> To avoid ambiguity and improve interpretability for the AI community, we have replaced the term “spatial resolution” with “beam footprint” throughout the paper and have added a clarifying explanation in Section 3.2. Additionally, we have corrected a previous inaccuracy: the beam footprint is 1–5 cm for a measurement range of 20–200 cm, rather than the ∼10 cm previously stated.
>
> **Q4: Could the different material plate sizes in Figure 2 introduce bias into the RF data across classes?**
>
> **Answer**: The varying material plate sizes do not introduce bias into the RF data across classes in our setup. This is because the FMCW beam in our system is focused to an effective footprint of approximately 1–5 cm at the center of each material plate, which ensures that all measurements are taken from a consistent sensing region independent of the plate’s overall dimensions. We have added corresponding clarifications in the revised manuscript Section 3.2. We have provided visualizations of the effective beam footprint on material plates in Appendix D.1 to further facilitate the AI community.
>
> **Reference:**
>
> [1] Chen, G., Luo, C., Zeng, H., Wen, G., Luo, Z., Wang, J., ... & Li, J. (2025). Material-ID: Towards mmWave-based Material Identification. ACM Transactions on Sensor Networks, 21(4), 1-26.
>
> [2] Wu, C., Zhang, F., Wang, B., & Liu, K. R. (2020). mSense: Towards mobile material sensing with a single millimeter-wave radio. Proceedings of the ACM on Interactive, Mobile, Wearable and Ubiquitous Technologies, 4(3), 1-20.
>
> [3] Huang, B., Gan, Y., Fang, F., Zhou, B., & Dai, X. (2024, July). Research on vibration suppression for industrial robots based on dynamic feedforward control. In 2024 43rd Chinese Control Conference (CCC) (pp. 4398-4403). IEEE.
>
> [4] Ito, K., & Iwasaki, M. (2016, October). State feedback-based vibration suppression for multi-axis industrial robot with posture change. In IECON 2016-42nd Annual Conference of the IEEE Industrial Electronics Society (pp. 5119-5124). IEEE.
>
> [5] Zhang, F., & Pan, S. (2013). Background-free millimeter-wave ultra-wideband signal generation based on a dual-parallel Mach-Zehnder modulator. Optics Express, 21(22), 27017-27022.

---

> ### Author Response · Authors · 2025-11-27
> **Looking forward to your reply**
>
> Dear Reviewer VGTy,
>
> As the rebuttal deadline is approaching, could you please have a look at our response? We are looking forward to your reply!
>
> Feel free to let us know if you have any other concerns. Thanks for your time and effort!
>
> Best Regards,
>
> Authors of Submission 9536

---

### Author Response · Authors · 2025-11-22
**Summary of the rebuttal and the major changes of revised manuscript**

Dear reviewers,

We would like to express our sincere gratitude for the valuable time, expertise, and thoughtful attention in reviewing our manuscript. The insightful comments, constructive feedback, and domain-specific suggestions have significantly strengthened the clarity, technical depth, and overall rigor of our work.

We are grateful that all reviewers acknowledge the contributions and advantages of RF-MatID: “**the first large-scale and open-source RF dataset covering a wide frequency range**” (Reviewer VGTy), a benchmark that is “**well motivated**” with “**large dataset, effective preprocessing, and comprehensive benchmark**” (Reviewer vQa1), a dataset that “**addresses a current gap in RF-based material sensing**” (Reviewer FgEZ),  and a resource that is “**clearly described**” with a “**well-structured taxonomy covering comprehensive materials**” (Reviewer QnNM).

We also recognized some concerns should be addressed to improve the manuscript. Therefore, we have diligently addressed all inquiries through extensive clarifications, enriched discussions, and additional experiments. **A revised manuscript** has been submitted, with all updates highlighted in magenta in both the main paper and the appendix. We provide here a summary of the significant changes implemented in the rebuttal and the revised manuscript:

**Clarification of Novelty and Scope for Embodied AI**: Added a clear explanation that RF-based material identification remains largely unexplored due to the absence of publicly accessible large-scale datasets and sensors capable of providing rich frequency-domain information. Clarified that RF-MatID is the first systematic dataset tailored for embodied AI scenarios rather than broad industrial automation.

**Terminology & Conceptual Precision**: Refined ambiguous terminology, clarified perturbation definitions, ensured acronym consistency, and improved the clarity of RF sensing concepts throughout the manuscript.

**AI-Centered Clarifications**: Enhanced the AI-inspired discussions and elaborations revealing RF-MatID’s contributions and impacts towards the AI community. Added a new appendix subsection elaborating AI-oriented intuitions grounded in material-specific mechanical properties.

**Custom Sensor Design Rationale**: Clarified the motivation and design of the custom, non–off-the-shelf wide-band RF sensor with a small beam footprint, developed to overcome the limitations of commercial devices and to enable fine-grained material sensing in indoor embodied-AI scenarios.

**Benchmark Expansion**: Incorporated new experiments on 4 additional RF-based material identification approaches, covering traditional methods and RF-tailored deep learning models, to provide a more comprehensive and informative benchmark.

**Dataset Limitations & Bias Discussion**: Added detailed analysis of dataset limitations, potential biases, representativeness, and cost/practicality of collection. Improved the physics-level explanation of system behavior and real-world considerations.

We believe these substantial revisions, together with the detailed rebuttal and enriched manuscript, have significantly improved the quality, clarity, and completeness of RF-MatID. We truly appreciate the reviewers’ efforts and constructive guidance, which have played a fundamental role in enhancing this work.

Best Regards,

Authors of Submission 9536

---

### Author Response · Authors · 2025-12-01
**Comprehensive Summary of Reviews and Author Rebuttal**

Dear Area Chair,

We sincerely appreciate your efforts to ensure a fair and rigorous review process, especially given the unusual circumstances surrounding the reassignment. To assist with your evaluation, we summarize the status of our submission below.

In the initial review before rebuttal, we appreciate that the reviewers acknowledged the contributions and advantages of our work: “**the first large-scale and open-source RF dataset covering a wide frequency range**” (Reviewer VGTy), a benchmark that is “**well motivated**” with “**large dataset, effective preprocessing, and comprehensive benchmark**” (Reviewer vQa1), a dataset that “**addresses a current gap in RF-based material sensing**” (Reviewer FgEZ),  and a resource that is “**clearly described**” with a “**well-structured taxonomy covering comprehensive materials**” (Reviewer QnNM).

We achieved an actual score profile of **(6, 6, 6, 4)** prior to the system rollback with only one reviewer response that increases the score from 2 to 6. On Nov 20, Reviewer QnNM (Score: 2$\to$6) confirmed that “*The responses have addressed most of my concerns*” and explicitly appreciated that “*Given the novelty of the work, the revision with corrected terms, and the comprehensiveness of the evaluation, I am willing to raise my scores accordingly.*” The remaining three reviewers (holding scores of 6, 6, and 4) have not yet responded to our detailed rebuttal.

**Summary of Contribution**

1. **A Gap in Robust Material Identification (Motivation)**: The paper identifies a critical limitation in material identification, where vision-based methods are inherently limited by optical ambiguity and the inability to capture the intrinsic physical properties for robust material identification. Although Radio Frequency (RF) sensing offers a solution to reveal the intrinsic material properties, its development is hindered by the lack of large-scale open-source datasets and information-rich wide-band hardware, preventing the systematic benchmarking of learning-based approaches.
2. **Physically Grounded Problem Formulation**: Grounded in electromagnetic principles showing that wave–matter interactions implicitly encode intrinsic physical properties into RF signal patterns, our work designs a customized wide-band high-precision sensor and systematically investigates how learning-based approaches can utilize these latent features to achieve robust, fine-grained material identification.
3. **Largest Scale and Comprehensive Dataset**: RF-MatID represents the largest and most comprehensive dataset for fine-grained material identification, covering 16 categories (5 superclasses) across a wide frequency band (4–43.5 GHz). It comprises 142k dual-domain samples (71k in each domain) and systematically incorporates geometric variations (incidence angle, stand-off distance).
4. **Extensive Benchmark**: RF-MatID introduces a rigorous benchmark setting by defining 5 frequency protocols (aligned with global regulations) and 7 data split settings (including out-of-distribution tests). We extensively evaluate 11 deep learning models (spanning CV, NLP, Time-Series, and RF research) across 3 protocols and 7 split settings. Additionally, a comprehensive baseline is constructed on all 5 protocols and 7 split settings.
5. **Impact on the Embodied AI Research Community**: By providing the first systematic open-source dataset and benchmark, RF-MatID enriches existing embodied perception frameworks by offering a complementary RF modality that, when fused with vision, provides fine-grained electromagnetic characteristics for comprehensive physical scene understanding, thereby enhancing the physical grounding capabilities of robots and automative machines VLA models. For instance, a robot can identify a glass cup's material pre-contact using a mounted RF sensor to enable adaptive grasping.

We kindly request the Area Chair to consider the critical gap our work addresses in RF-based material identification for embodied perception research. The substantial revisions, together with the expanded discussions and additional experiments, have considerably strengthened the clarity, rigor, and completeness of the manuscript. We hope that RF-MatID will contribute meaningfully to the ICLR readership and serve as a foundational large-scale dataset and comprehensive benchmark, empowering further material identification research in the community of physical AI to advance material-guided robotic manipulation and functional interactions.

Best regards,

Authors of Submission 9536

---

### Meta-Review · Area_Chair_EXhH · 2026-01-01

**Summary:**

This paper introduces RF-MatID, a novel, large-scale, open-source benchmark dataset designed for radio-frequency-based material identification. Addressing critical limitations in vision-only methods that struggle with optical ambiguity, this work leverages wide-band RF sensing to capture intrinsic physical material properties. The submission offers a comprehensive resource covering 16 material categories across a broad frequency range (4–43.5 GHz) and establishes a rigorous benchmark by evaluating numerous deep learning models across various protocols and data splits.

The review process highlighted the significant value of this contribution, with reviewers initially recognizing it as a well-motivated, first-of-its-kind large-scale dataset that fills a current gap in RF-based sensing. The authors provided a detailed rebuttal that successfully addressed major concerns, notably convincing the most critical reviewer to raise their score significantly based on the revisions and comprehensive evaluation. Despite some reviewers not responding to the final rebuttal, the overall positive reception and the clear utility of this foundational dataset for the embodied AI community support the decision to accept.

**Reviewer Concerns:**

The reviewers’ main concerns centered on

* Novelty & scope not crystal-clear (esp. for embodied AI): Clarified that RF-based material ID has been underexplored mainly due to the lack of large-scale public datasets and wide-band sensors, and positioned RF-MatID explicitly as a dataset designed for embodied indoor AI rather than general industrial automation.

* Ambiguous terminology / conceptual imprecision: Tightened definitions (e.g., “perturbation”), improved RF sensing explanations, standardized acronyms, and removed/rewrote unclear phrasing to reduce reader confusion.

* AI relevance under-emphasized: Expanded AI-centered discussion of why the dataset matters to the AI community and added an appendix subsection with AI-oriented intuitions tied to material-specific mechanical/physical properties.

* Sensor choice/design rationale insufficient: Added a clearer motivation for the custom wide-band, small-footprint RF sensor and explained why off-the-shelf devices are inadequate for fine-grained material sensing in embodied-AI indoor settings.

* Benchmark coverage too narrow: Added experiments for 4 additional RF material-ID methods, spanning traditional baselines and RF-tailored deep learning approaches, to make the benchmark more comprehensive and informative.

* Dataset limitations/bias not discussed enough: Added detailed analysis of limitations, potential biases, representativeness, and collection cost/practicality, plus strengthened physics-level explanations of system behavior and real-world constraints.

**Reviewer Scores:**

Reviewer QnNM raised from 2 to 6;

Reviewer FgEZ would raise score from 4 to 6

---

### Decision · Program_Chairs · 2026-01-26

Accept (Poster)